# PAGE-4D: VGGT-4D PERCEPTION VIA DISENTANGLED POSE AND GEOMETRY ESTIMATION

**Kaichen Zhou**[1,2] **Yuhan Wang**[2,3*] **Grace Chen**[1*] **Gaspard Beaudouin**[1,4]
**Fangneng Zhan**[2] **Paul Pu Liang**[2†] **Mengyu Wang**[1,5†]
[1]Harvard AI and Robotics Lab, Harvard University
[2]Media Lab and Electrical Engineering and Computer Science, Massachusetts Institute of Technology
[3]Department of Computing, Imperial College London
[4]École Nationale des Ponts et Chaussées, Institut Polytechnique de Paris
[5]Kempner Institute for the Study of Natural and Artificial Intelligence, Harvard University

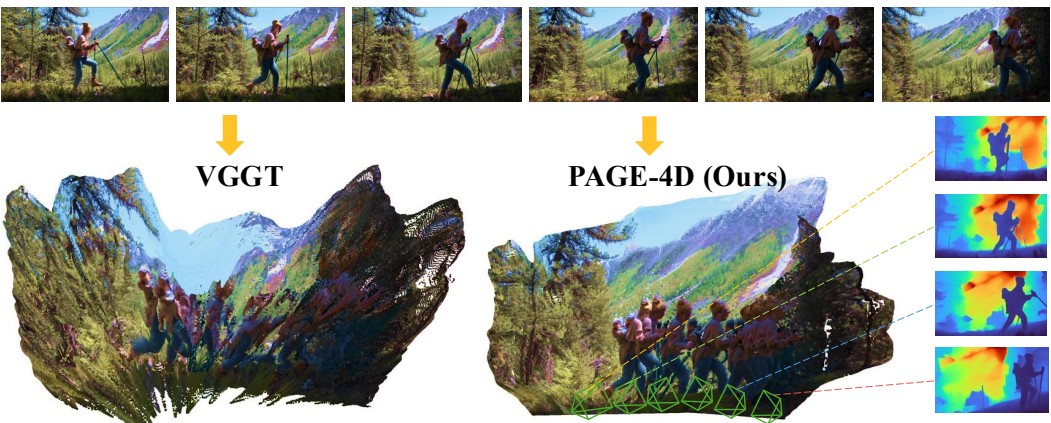

Figure 1: **PAGE-4D** takes a sequence of RGB images depicting a dynamic scene as input and simultaneously predicts the corresponding camera parameters and 3D geometry information—all within a fraction of a second. Compared to VGGT, **PAGE-4D** produces denser and more accurate point cloud reconstructions with better depth estimation quality. (Best viewed in PDF.)

## ABSTRACT

Recent 3D feed-forward models, such as the Visual Geometry Grounded Transformer (VGGT), have shown strong capability in inferring 3D attributes of static scenes. However, since they are typically trained on static datasets, these models often struggle in real-world scenarios involving complex dynamic elements, such as moving humans or deformable objects like umbrellas. To address this limitation, we introduce PAGE-4D, a feedforward model that extends VGGT to dynamic scenes, enabling camera pose estimation, depth prediction and point cloud reconstruction —all without post-processing. A central challenge in multi-task 4D reconstruction is the inherent conflict between tasks: accurate camera pose estimation requires suppressing dynamic regions, while geometry reconstruction requires modeling them. To resolve this tension, we propose a dynamics-aware aggregator that disentangles static and dynamic information by predicting a dynamics-aware mask—suppressing motion cues for pose estimation while amplifying them for geometry reconstruction. Extensive experiments show that PAGE-4D consistently outperforms the original VGGT in dynamic scenarios, achieving superior results in camera pose estimation, monocular and video depth estima-

---

*These authors contributed equally as the second authors. †These authors jointly supervised this work.
The acknowledgments for the video used in Fig. 1 and Fig. 5 are provided in the appendix.

tion, and dense point map reconstruction. Necessary code and additional demos are available at `Link`.

**Keywords**: VGGT-4D, 4D Perception, Dynamic Scene Reconstruction.

# 1 INTRODUCTION

Despite recent advances in feedforward 3D estimation of static scenes from image sets (Zhang et al., 2025a; Wang et al., 2025a; 2024), extending these capabilities to dynamic environments—scenes where objects or people undergo motion or deformation— remains a significant challenge due to the complexity of real-world motion. A common strategy for handling dynamic scenarios is to decompose the problem into a series of sub-modules, such as depth estimation, optical flow computation, and object tracking (Luiten et al., 2020; Mustafa et al., 2016; Kopf et al., 2021; Zhang et al., 2022b). While this modular approach simplifies the task by disentangling different components, it often results in increased computational cost and error accumulation across sequential stages (Zhang et al., 2025b). Given the limitations of modular pipelines, a unified method for dynamic geometry learning that avoids sequential decomposition offers a more effective and coherent solution. However, developing such models usually requires capturing spatiotemporal relationships across frames, and demands notable computational resources as well as access to large-scale dynamic datasets with ground-truth geometry (Zhang et al., 2025b; Wang et al., 2025b).

Motivated by these challenges, we present **PAGE-4D**, a unified and efficient feed-forward model that enables the inference of key 3D attributes in dynamic scenes as shown in Fig. 1. To address the limited availability of labeled dynamic data, we build on the pretrained 3D foundation model VGGT (Wang et al., 2025a) and adapt it to dynamic scenarios through targeted fine-tuning. While VGGT demonstrates strong performance in static scene understanding, its accuracy drops significantly when applied to dynamic environments involving people, vehicles, or deformable objects. This limitation stems from a fundamental tension: motion provides valuable cues for geometry estimation in dynamic scenarios, yet simultaneously introduces noise that corrupts camera pose estimation by violating the static epipolar constraint, as shown in Fig. 2 (a). In other words, the very signals that enable reconstructing dynamic objects are also those that hinder reliable pose recovery (Chen et al., 2025; Zhang et al., 2025b; 2022b).

This insight motivates our central idea: rather than viewing dynamics as uniformly harmful or helpful, we disentangle their effects across tasks. We introduce a dynamics-aware aggregator that first predicts a mask to identify dynamic regions, and then applies it via a cross-attention mechanism—filtering dynamic content for camera pose tokens while emphasizing it for geometry tokens. Together with targeted fine-tuning of layers most sensitive to dynamics, this design allows us to harness motion where it benefits geometry grounding, while suppressing its negative impact on pose estimation. With this design, our method achieves accurate pose and geometry estimation for both static and dynamic content in challenging dynamic scenarios, as illustrated in Fig. 1. Through extensive experiments, PAGE-4D establishes new state-of-the-art performance across multiple benchmarks and tasks. For instance, on the Sintel benchmark, it reduces the camera pose estimation ATE from 0.214 (VGGT) to 0.143 and improves the scale-aligned video depth Abs Rel from 0.484 to 0.357. Notably, thanks to its plug-in design, PAGE-4D adds only a negligible overhead in both runtime and storage compared to VGGT. This work makes the following key contributions:

- We propose PAGE-4D, a dynamic-aware extension of VGGT for 4D scene understanding, which achieves state-of-the-art results on dynamic geometry perception benchmarks.

- We design a dynamics-aware aggregator that combines (i) a mask prediction module for identifying dynamic regions and (ii) a global attention mechanism that selectively leverages or suppresses dynamic information across tasks.

- We provide an in-depth analysis of VGGT under dynamic conditions and introduce a targeted fine-tuning strategy that adapts only the layers most sensitive to dynamics, enabling efficient transfer by updating only a limited subset of parameters.

## 2 RELATED WORK

**3D Feedforward Model** is learning-based approach that reconstructs static 3D scene geometry from input images with temporal invariance assumption (Bochkovskii et al., 2024; Yin et al., 2023; Piccinelli et al., 2024; Leroy et al., 2024), treating all views as capturing the same static scene. DUSt3R (Wang et al., 2024) is the representative of this reconstruction framework, introducing transformer-based architectures that processes image pairs from different viewpoints, learning direct mappings from 2D image pixels to 3D coordinate fields. Subsequent works (Tang et al., 2024; Yang et al., 2025; Wang et al., 2025b; Bhat et al., 2023; Tang & Tan, 2018; Yao et al., 2018; Chen et al., 2021) have explored broader scenarios. Among those, VGGT (Wang et al., 2025a) presents a unified architecture using alternating attention mechanisms within each frame and across the entire sequence, responding the need of joint prediction of camera poses, depth maps, and point correspondences through integrated training. Despite these advances, traditional 3D methods remain temporally invariant and struggle with dynamic scenes, motivating the need for 4D feedforward approaches that explicitly capture scene dynamics.

**4D Feedforward Model** emerges to reconstruct dynamic scenes by capturing geometric evolution over time from image sequences (Tian et al., 2023; Van Hoorick et al., 2022; Büsching et al., 2024; Liang et al., 2024; Zhao et al., 2023). However, it faces significant challenges in modeling temporal geometry changes, as moving objects violate the rigid geometry assumptions of static methods (Oliensis, 2000; Ozyesil et al., 2017; Cao et al., 2025). Given DUSt3R's success in static reconstruction, several works (Lu et al., 2024; Wu et al., 2025; Wang & Agapito, 2024; Xu et al., 2024; Yao et al., 2025) have adapted this framework for dynamic scenarios. While MONST3R (Zhang et al., 2025b) fine tunes DUSt3R on video sequences, D²USt3R (Han et al., 2025) introduces explicit temporal modeling through 4D pointmap representations and cross-frame attention mechanisms, improving in establishment of correspondences between moving objects across frames. Other efforts include training-free methods like Easi3R (Chen et al., 2025). Despite the progress shown by these DUSt3R-based approaches in dynamic content, they are all constrained by the pairwise progressing framework in DUSt3R. Alternative approaches (Feng et al., 2025; Li et al., 2024; Xu et al., 2025; Jiang et al., 2025; Jin et al., 2025; Piccinelli et al., 2024; Bochkovskii et al., 2024) explore different architectural designs to handle dynamic scenes for task-specific solutions, but they often sacrifice the generalizability of feedforward approaches. More recently, the success of VGGT in sequence-based static reconstruction has inspired its extensions (Li et al., 2025) to dynamic scenarios. Recent approaches such as MoVieS (Lin et al., 2025) and StreamVGGT (Zhuo et al., 2025) focus on narrow, application-specific scenarios rather than the broader challenge of adapting static models to dynamic domains. In contrast, we propose PAGE-4D to address this general challenge, demonstrating that carefully targeted fine-tuning of key attention components can effectively bridge the static–dynamic divide without requiring major architectural changes.

## 3 METHODOLOGY

In this paper, we extend the VGGT to PAGE-4D (Disentangled Pose and Geometry Estimation for 4D Perception), a dynamic-aware framework for robust 4D scene understanding. Given a sequence of $N$ RGB frames $\{\mathbf{I}_i\}_{i=1}^N$ captured in a dynamic environment, our objective is to predict temporally consistent 3D outputs for each frame:

$$f\big(\{\mathbf{I}_i\}\big) = \big\{(\mathbf{g}_i, \mathbf{D}_i, \mathbf{P}_i\big\}_{i=1}^N,$$

where $\mathbf{g}_i \in \mathbb{R}^9$ encodes the camera intrinsics and extrinsics, $\mathbf{D}_i \in \mathbb{R}^{H \times W}$ is the depth map, and $\mathbf{P}_i$ the 3D point map.

In this section, we begin by examining the behavior of VGGT under dynamic conditions and analyzing how its transformer architecture represents spatiotemporal information. This analysis reveals fundamental limitations when directly applying VGGT to dynamic scenes. Guided by these insights, we introduce PAGE-4D, a principled yet lightweight extension of VGGT that enables accurate and efficient estimation of camera pose, geometry, and tracking in challenging dynamic environments.

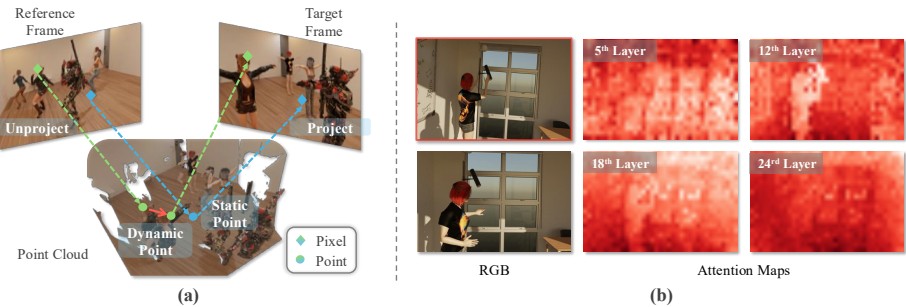

Figure 2: **Motivating illustration:** (a) In static scenes, geometric consistency is preserved across frames, while in dynamic scenes, moving objects violate this consistency. (b) Visualization of VGGT attention maps from the 5th, 12th, 18th, and 24th layers of global attention block with the method in Caron et al. (2021). Attention values are visualized using a white-to-red color map, with white indicating low values and red indicating high values. VGGT tends to ignore dynamic content during the feed-forward process, which motivates our design of the dynamics-aware mask.

## 3.1 MOTIVATION

**Empirical Observation:** Although VGGT achieves state-of-the-art performance in static scene understanding, its accuracy degrades markedly in the presence of dynamic objects. On the Odyssey test set (Zheng et al., 2023), which evaluates long-range point tracking and geometry understanding in dynamic scenes, we directly apply VGGT for evaluation. The results reveal a clear gap between static and dynamic regions: the Absolute Depth Error in dynamic regions is 94% higher than in static regions. These results highlight the need for an architecture that achieves reliable scene understanding across both static and dynamic scenarios.

To better understand this gap, we first follow Chefer et al. (2021) on feature visualization and examine key layers of VGGT (Fig. 2 (b)). We observe that dynamic regions exhibit weaker activations compared to static ones, suggesting that VGGT tends to ignore dynamic content. We then perform an ablation in which attention among dynamic tokens is explicitly suppressed (see Appendix). Masking dynamic patches from the cross-frame attention mechanism improves camera pose estimation, but at the same time leads to a sharp drop in geometry. Together, these findings reveal a fundamental tension in dynamic scenes: *while camera pose estimation benefits from suppressing dynamic regions to maintain epipolar consistency, geometry requires exploiting their motion cues*.

**Static Case – Geometric Foundations:** Formally, under static conditions, geometry estimation can be achieved by implicitly modeling the correspondence between a reference-frame homogeneous pixel $\mathbf{x}_r$ and its target-frame homogeneous pixel $\mathbf{x}_t$ (Fig. 2 (a)) (Zhang et al., 2025b; Chen et al., 2025), which is fully determined by the camera intrinsics, depth, and relative pose:

$$\mathbf{x}_t = \mathbf{K} \left[ \mathbf{R}_{t \leftarrow r} \; D_r(\mathbf{x}_r) \; \mathbf{K}^{-1} \; \mathbf{x}_r + \mathbf{t}_{t \leftarrow r} \right], \tag{1}$$

This equation encodes the standard rigid-scene geometry assumption: once depth and camera motion are known, pixel correspondences across frames can be predicted without ambiguity. Meanwhile, pose estimation (Zhang et al., 2025b; Chen et al., 2025), due to the concentration of relative camera motion, in VGGT often reduces to fitting an essential matrix $\mathbf{E}$ that enforces the epipolar constraint between normalized homogeneous pixels $\tilde{\mathbf{x}}_r$ and $\tilde{\mathbf{x}}_t$:

$$\tilde{\mathbf{x}}_t^\top \; \mathbf{E} \; \tilde{\mathbf{x}}_r = 0, \quad \mathbf{E} = [\, \mathbf{t}_{t \leftarrow r} \,]_\times \; \mathbf{R}_{t \leftarrow r}. \tag{2}$$

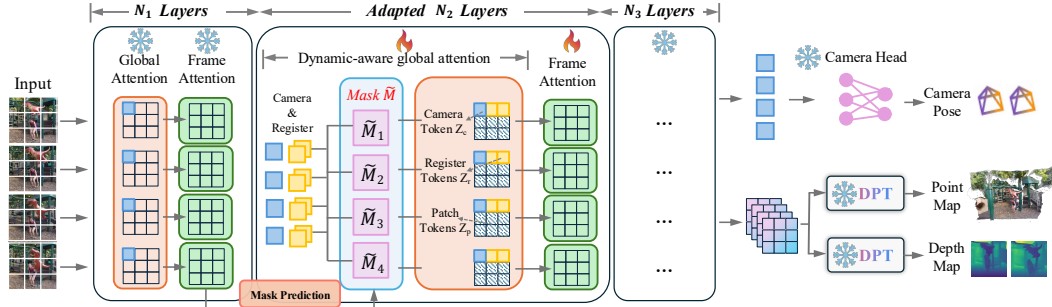

Figure 3: **Fine-tuning strategy:** Instead of fine-tuning the entire VGGT architecture, we adapt only the middle $N_2$ layers of the global attention mechanism, which are most critical for cross-frame information fusion. To further address dynamic scenes, we introduce a *dynamics-aware aggregator* that predicts a mask to disentangle dynamic and static content.

*Summary:* Under static conditions, both Eqn. 1 and Eqn. 2 hold for all non-occluded pixel pairs $(\mathbf{x}_r, \mathbf{x}_t)$ between the reference and target frames. This makes the joint optimization of camera tokens and geometry tokens over the same patches within a frame a reasonable design choice.

**Dynamic Case – Violation and Residuals:** In dynamic scenes, to realize geometry estimation, motion information needs to be taken into consideration for accurate prediction:

$$\mathbf{x}_t = \mathbf{K} \left[ \mathbf{R}_{t \leftarrow r} \, D_r(\mathbf{x}_r) \, \mathbf{K}^{-1} \mathbf{x}_r + \mathbf{t}_{t \leftarrow r} \right] + \mathbf{K} \mathbf{M}_{t \leftarrow r}, \qquad (3)$$

where $\mathbf{M}_{t \leftarrow r}$ represents the displacement induced by object motion. Meanwhile, in the presence of dynamic motion, the Eqn. 2 for pose estimation no longer holds. The violation manifests as a residual:

$$\delta(\mathbf{x}_r) \equiv \tilde{\mathbf{x}}_t^\top \mathbf{E} \tilde{\mathbf{x}}_r \quad \approx \quad \frac{1}{Z_r} \, \mathbf{n}(\mathbf{x}_r)^\top \Delta \mathbf{X}_\perp(\mathbf{x}_r), \qquad (4)$$

where $\mathbf{n}(\mathbf{x}_r)$ is the unit normal of the epipolar line associated with $\mathbf{x}_r$, and $\Delta \mathbf{X}_\perp(\mathbf{x}_r)$ is the component of the dynamic displacement perpendicular to that line. This residual quantifies the degree to which dynamic motion "pushes" correspondences away from the epipolar geometry predicted by the camera. The larger the residual, the stronger the violation of the static-scene assumption, and the greater the resulting pose estimation error. Eqn. 4 implies that in dynamic scenarios, only the static subset of pixel pairs $(\mathbf{x}_r^{sta}, \mathbf{x}_t^{sta})$ satisfy Eqn. 2.

*Summary:* Under dynamic conditions, Eqn. 3 for geometry estimation remains valid for all non-occluded pixel pairs $(\mathbf{x}_r, \mathbf{x}_t)$ between the target and reference frames, whereas Eqn. 2 for pose estimation holds only for the static subset of non-occluded pixels $(\mathbf{x}_r^{sta}, \mathbf{x}_t^{sta})$, as explained in Eqn. 4.

*Insight:* Under dynamic scenarios, camera pose estimation is brittle to dynamic motion, as small residuals can corrupt essential matrix fitting, while geometry and tracking tasks can in fact benefit from modeling $\mathbf{M}_{t \leftarrow r}$. Motivated by this insight, we propose PAGE-4D, a dynamic-aware extension of VGGT that disentangles the role of dynamic regions across tasks—suppressing them for pose estimation while leveraging them for geometry and tracking.

### 3.2 PAGE-4D

PAGE-4D is composed of four key components: (1) a pre-trained DINO-style (Zhang et al., 2022a) encoder that extracts image-level representations; (2) a *dynamics-aware aggregator* that integrates spatial and temporal cues through three modules—Frame Attention for inter-frame patch relations, Global Attention for intra-frame patch relations, and Dynamics-Aware Global Attention for disentangling dynamic from static content; (3) lightweight decoders for depth, 3D point maps; and (4) a larger decoder dedicated to camera pose estimation.

PAGE-4D inherits components (1), (3), and (4) directly from VGGT, while extending component (2) into a three-stage dynamics-aware aggregator as in Fig. 3. The first stage consists of $N_1$ layers,

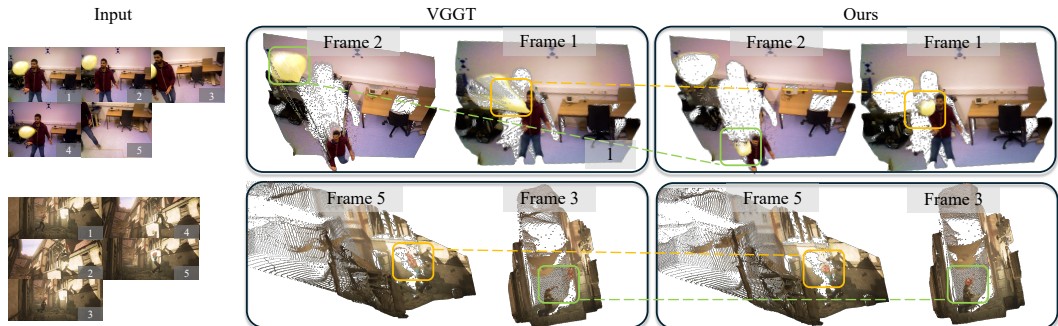

Figure 4: **Qualitative Comparison of Point Cloud Estimation on the Bonn & Sintel**: As shown in the figure, our method effectively captures the geometric structure in scenarios with complex motion, whereas VGGT produces fragmented and inconsistent geometry. (Best viewed in PDF.)

each composed of one Global Attention and one Frame Attention block. Its output is fed into a dynamic mask prediction module, which produces a dynamics-aware mask. This mask is then applied in the second stage to disentangle dynamic and static content for pose and geometry estimation. The second stage itself consists of $N_2$ layers, each comprising a Dynamics-Aware Global Attention block and a Frame Attention block. The final stage consists of $N_3$ layers as the first stage.

### 3.2.1 Dynamics Mask Prediction

A central challenge in dynamic scenes is to selectively suppress the influence of moving objects for tasks such as pose estimation, while still retaining their information for geometry. To achieve this, we design a dynamic mask prediction module that learns, in a self-supervised manner, which spatial regions are likely to correspond to dynamic objects. This is feasible because the middle layers of PAGE-4D already disentangle dynamic and static content, as illustrated in Fig. 2(b), where dynamic regions are treated distinctly. Formally, given the token features $\mathbf{z} \in \mathbb{R}^{B \times S \times P \times d}$ from the aggregator, we first extract only the patch tokens $\mathbf{z}_p \in \mathbb{R}^{B \times S \times (H_P \cdot W_P) \times d}$. These tokens are projected into a lower-dimensional representation via a linear mapping, followed by a depthwise convolutional head that produces mask logits: $\mathbf{m} = \mathrm{Conv}(\mathbf{z}_p) \in \mathbb{R}^{(B \cdot S) \times 1 \times H_P \times W_P}$.

### 3.2.2 Mask Attention

Once the dynamic mask $\widetilde{\mathbf{M}}$ has been predicted, it can be directly incorporated into the transformer attention mechanism. Specifically, given queries $\mathbf{Q}$, keys $\mathbf{K}$, and values $\mathbf{V}$, attention is:

$$\mathrm{Attn}(\mathbf{Q}, \mathbf{K}, \mathbf{V}) = \mathrm{softmax}\left( \frac{\mathbf{Q}\mathbf{K}^\top}{\sqrt{d}} + \widetilde{\mathbf{M}} \right) \mathbf{V}, \tag{5}$$

where $\widetilde{\mathbf{M}} \in \mathbb{R}^{B \times (S \cdot P) \times (S \cdot P)}$ is the broadcasted mask applied to the attention logits. Importantly, we apply this mask in a task-specific manner: **Pose estimation.** For queries corresponding to the camera token and registration token, $\widetilde{\mathbf{M}}$ actively suppresses attention to dynamic regions, ensuring that pose estimation remains consistent with epipolar geometry and static scene constraints. **Depth and Point Cloud.** For patches concerning these tasks, the mask is not applied, allowing the network to leverage dynamic motion cues to improve point map reconstruction and 2D–3D tracking accuracy.

This asymmetric design explicitly disentangles the role of dynamic regions across tasks. Dynamic objects, which are detrimental for camera pose estimation, are ignored in that context, but their motion signals remain available for geometry and tracking tasks. By learning the mask in a fully differentiable manner, the model adapts its behavior to the motion patterns present in the training data, rather than relying on pre-defined heuristics.

**Memory-Efficient Mask Mechanism** Although Eq. 5 describes a full $(S \cdot P)^2$ mask, forming this matrix would require $\mathcal{O}(N^2)$ memory and break fused Scaled Dot-Product Attention, where $N = S \cdot P$. PAGE-4D instead implements an *equivalent additive mask* using two vectors. Given attention

inputs $\mathbf{Q}, \mathbf{K}, \mathbf{V} \in \mathbb{R}^{N \times d}$, the mask head predicts:

$$r \in \mathbb{R}^N, \qquad c \in \mathbb{R}^N.$$

We append these to the feature dimension:

$$q'_i = [\, q_i \sqrt{d'/d}, \; r_i \sqrt{d'} \,], \qquad k'_j = [\, k_j, \; c_j \,], \qquad v'_j = [\, v_j, \; 0 \,],$$

where $d' = d + 1$. Then:

$$\frac{q'_i k'^{\top}_j}{\sqrt{d'}} = \frac{q_i^{\top} k_j}{\sqrt{d}} + r_i c_j,$$

but *without constructing* the $N \times N$ mask.

This uses only $\mathcal{O}(N)$ memory, stays compatible with fused Scaled Dot-Product Attention.

### 3.3 TRAINING DETAILS

**Fine-tuning Strategy.** During fine-tuning, we update only the middle 10 layers while freezing the remaining aggregator and decoder layers, thereby tuning just 30% of the model instead of the full network. This design is supported by studies on transformer representations, which show that lower layers capture local structures, middle layers model regional relationships, and higher layers encode global semantics (Raghu et al., 2021; Caron et al., 2021). Moreover, as illustrated in Fig 2(b), the middle layers of VGGT tend to suppress dynamic content, leading to degraded performance in dynamic scenarios. By selectively fine-tuning these layers, we aim to reintroduce dynamic information into the feed-forward process. Consistent with this intuition, our ablations (Please Refer to Appendix) confirm that the later middle layers contribute most significantly to accurate geometry estimation.

**Loss Functions.** We adopt a multi-task loss combining supervision for camera pose, depth and point-maps:

$$\mathcal{L} = \lambda_c \mathcal{L}_{\text{camera}} + \mathcal{L}_{\text{depth}} + \mathcal{L}_{\text{pmap}}. \tag{6}$$

Following VGGT, we empirically set the loss weights to balance gradients across tasks, with $\lambda_c = 5$. We adopt: Huber loss for camera pose estimation, Uncertainty-weighted depth and point-map losses with gradient regularization. We do not include point tracking in our model, since the tracking head in VGGT is primarily designed for view registration and is not well-suited to dynamic scenarios. In addition, VGGT does not provide clear training code for the tracking head. These two factors prevent us from incorporating point tracking into our framework.

## 4 EXPERIMENTS

To evaluate the effectiveness of PAGE-4D, we apply it to monocular video sequences and assess its performance on five tasks: video depth estimation, monocular depth estimation, camera pose estimation, multi-view point map reconstruction, and 4D view synthesis. We compare against several strong baselines—DUSt3R (Wang et al., 2024), MASt3R (Leroy et al., 2024), MonST3R (Zhang et al., 2025b), CUT3R (Wang et al., 2025b), Fast3R (Yang et al., 2025), FLARE (Zhang et al., 2025c), and VGGT (Wang et al., 2025a)—across each subtask.

### 4.1 VIDEO DEPTH ESTIMATION

Following the protocol of prior works (Wang et al., 2024; Zhang et al., 2025b), we evaluate our approach on the video depth estimation task using Sintel (Butler et al., 2012) and Bonn (Palazzolo et al., 2019). To assess robustness to dynamic objects, we additionally incorporate DyCheck (Gao et al., 2022). We report Absolute Relative Error (Abs Rel) and prediction accuracy at the threshold $\delta < 1.25$, under two alignment settings: (i) scale-only alignment and (ii) joint scale and 3D translation alignment. Qualitative results could be found in Fig 4 and Fig 5. As summarized in Tab. 1, our method establishes a new state of the art across all three datasets and both alignment settings among feed-forward 3D reconstruction models. Compared to VGGT (Wang et al., 2025a), which represents the strongest prior baseline, our approach consistently reduces error and improves accuracy. For example, on Sintel with scale-shift alignment, we improve $\delta < 1.25$ accuracy from 0.639

Table 1: **Video Depth Estimation on Sintel (Butler et al., 2012), Bonn (Palazzolo et al., 2019) and DyCheck (Yang et al., 2025).** FPS is evaluated on KITTI using one A800 GPU. Missing entries (–) denote results not reported in the original papers cited.

| Method | Params | Align | Sintel | | Bonn | | DyCheck | | FPS |
|---|---|---|---|---|---|---|---|---|---|
| | | | Abs Rel ↓ | $\delta < 1.25$ ↑ | Abs Rel ↓ | $\delta < 1.25$ ↑ | Abs Rel ↓ | $\delta < 1.25$ ↑ | |
| DUSt3R (Wang et al., 2024) | 571M | | 0.662 | 0.434 | 0.151 | 0.839 | - | - | 1.25 |
| MASt3R (Leroy et al., 2024) | 689M | | 0.558 | 0.487 | 0.188 | 0.765 | - | - | 1.01 |
| CUT3R (Wang et al., 2025b) | 793M | scale Video Depth | 0.430 | 0.465 | **0.077** | **0.937** | 0.176 | 0.740 | 6.98 |
| Fast3R (Yang et al., 2025) | 648M | | 0.638 | 0.422 | 0.194 | 0.772 | - | - | 65.8 |
| FLARE (Zhang et al., 2025c) | 1.40B | | 0.729 | 0.336 | 0.152 | 0.790 | - | - | 1.75 |
| VGGT (Wang et al., 2025a) | 1.26B | | 0.484 | 0.553 | 0.107 | 0.883 | 0.182 | 0.743 | 43.2 |
| **PAGE-4D(Ours)** | 1.26B | | **0.357** | **0.699** | 0.092 | 0.904 | **0.170** | **0.785** | 43.2 |
| DUSt3R (Wang et al., 2024) | 571M | | 0.570 | 0.493 | 0.152 | 0.835 | - | - | 1.25 |
| MASt3R (Leroy et al., 2024) | 689M | | 0.480 | 0.517 | 0.189 | 0.771 | - | - | 1.01 |
| CUT3R (Wang et al., 2025b) | 793M | scale&shift Video Depth | 0.534 | 0.558 | **0.075** | **0.943** | 0.228 | 0.635 | 6.98 |
| Fast3R (Yang et al., 2025) | 648M | | 0.518 | 0.486 | 0.196 | 0.768 | - | - | 65.8 |
| FLARE (Zhang et al., 2025c) | 1.40B | | 0.791 | 0.358 | 0.142 | 0.797 | - | - | 1.75 |
| VGGT (Wang et al., 2025a) | 1.26B | | 0.261 | 0.639 | 0.102 | 0.890 | 0.155 | 0.792 | 43.2 |
| **PAGE-4D(Ours)** | 1.26B | | **0.212** | **0.763** | 0.090 | 0.903 | **0.145** | **0.854** | 43.2 |
| DUSt3R (Wang et al., 2024) | 571M | | 0.488 | 0.532 | 0.139 | 0.832 | - | - | 1.25 |
| MASt3R (Leroy et al., 2024) | 689M | | 0.413 | 0.569 | 0.123 | 0.833 | - | - | 1.01 |
| MonST3R (Zhang et al., 2025b) | 571M | | 0.402 | 0.525 | 0.069 | 0.954 | - | - | 1.27 |
| CUT3R (Wang et al., 2025b) | 793M | Monocular Depth | 0.418 | 0.520 | 0.058 | 0.967 | 0.149 | 0.790 | 6.98 |
| Fast3R (Yang et al., 2025) | 648M | | 0.544 | 0.509 | 0.169 | 0.796 | - | - | 65.8 |
| FLARE (Zhang et al., 2025c) | 1.40B | | 0.606 | 0.402 | 0.130 | 0.836 | - | - | 1.75 |
| VGGT (Wang et al., 2025a) | 1.26B | | 0.292 | 0.629 | 0.071 | 0.947 | 0.160 | 0.799 | 43.2 |
| **PAGE-4D(Ours)** | 1.26B | | **0.242** | **0.742** | **0.053** | **0.970** | **0.141** | **0.840** | 43.2 |

VGGT to 0.763 (+19.4%) while lowering Abs Rel from 0.261 to 0.212 (-18.8%). Similar trends are observed on Sintel and Bonn, where our method outperforms VGGT under both alignment regimes, without incurring noticeable increases in speed or memory consumption.

## 4.2 MONOCULAR DEPTH ESTIMATION

In addition to video depth, we evaluate our approach on monocular depth estimation following Leroy et al. (2024); Zhou et al. (2025). Each predicted depth map is aligned independently with its ground truth, in contrast to the video setting where a single scale (and shift) is applied across the entire sequence. As summarized in Tab. 1, our method shows consistent improvements over existing feed-forward reconstruction methods. In particular, compared to VGGT (Wang et al., 2025a) in Sintel dataset, our approach reduces Abs Rel from 0.292 to 0.242, and increases $\delta < 1.25$ accuracy from 0.629 to 0.742. While not explicitly optimized for single-frame depth estimation, our method performs favorably against dedicated baselines such as DUSt3R, MONST3R, and FLARE. These results suggest that our model generalizes well from video sequences to single-image inputs.

## 4.3 CAMERA POSE ESTIMATION

We evaluate camera pose estimation on the dynamic-scene Sintel (Butler et al., 2012) and Tum (Sturm et al., 2012) benchmarks. Following the protocol in (Zhang et al., 2025b), we report Absolute Tra-

Table 2: **Camera Pose Estimation on Sintel and Tum.**

| Method | Optim. | Sintel | | | Tum | | |
|---|---|---|---|---|---|---|---|
| | | ATE ↓ | $RPE_{trans}$ ↓ | $RPE_{rot}$ ↓ | ATE ↓ | $RPE_{trans}$ ↓ | $RPE_{rot}$ ↓ |
| MonST3R (Zhang et al., 2025b) | ● | **0.108** | **0.042** | 0.732 | 0.098 | 0.019 | 0.935 |
| DUSt3R (Wang et al., 2024) | | 0.417 | 0.250 | 5.796 | 0.140 | 0.106 | 3.286 |
| Spann3R (Wang & Agapito, 2024) | | 0.329 | 0.110 | 4.471 | 0.056 | 0.021 | 0.591 |
| CUT3R (Wang et al., 2025b) | | 0.213 | 0.066 | 0.621 | 0.046 | 0.015 | 0.473 |
| VGGT (Wang et al., 2025a) | | 0.214 | 0.079 | 0.643 | 0.028 | 0.014 | 0.371 |
| **PAGE-4D(Ours)** | | 0.178 | 0.069 | **0.547** | **0.016** | **0.011** | **0.323** |

jectory Error (ATE), Relative Pose Error in translation ($RPE_{trans}$), and rotation ($RPE_{rot}$). For a fair comparison, predicted trajectories are aligned to the ground truth via Sim(3) Umeyama alignment, and we uniformly sample 10 frames per sequence for evaluation. As shown in Tab. 2, our method delivers substantial improvements on Tum, reducing $RPE_{trans}$ by 21% and $RPE_{rot}$ by 13% compared to prior feed-forward approaches, while maintaining competitive ATE. On Sintel, our approach also reduces $RPE_{rot}$ by 17%, highlighting its robustness across both synthetic and real-world dynamic scenes.

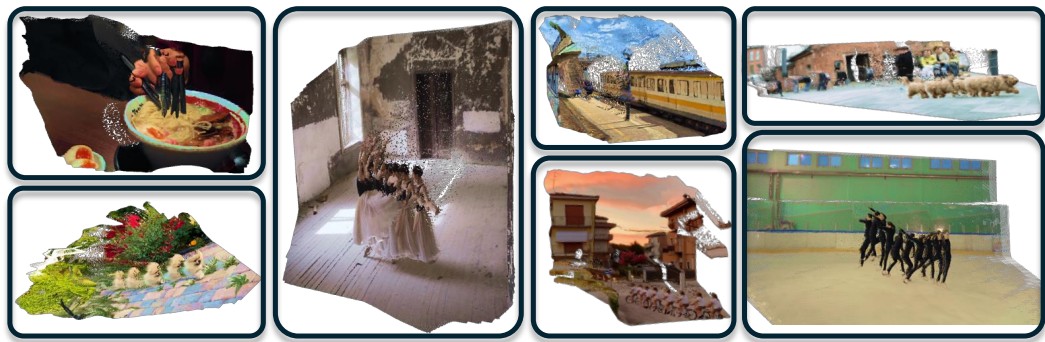

Figure 5: **Qualitative Results of Point Cloud Estimation.** PAGE-4D can estimate camera poses and depth maps from RGB inputs, even in the presence of dynamic objects. (Best viewed in PDF.)

## 4.4 POINT MAP ESTIMATION

We further evaluate our method on the DyCheck (Gao et al., 2022) benchmark for dynamic-scene point map reconstruction. Following the protocol of (Wang et al., 2024; Zhang et al., 2025b), we report Accuracy (Acc.),

Table 3: **Point reconstruction on DyCheck.**

| Method | Optim. | Acc ↓ | | Comp ↓ | | Overall ↓ | |
|---|---|---|---|---|---|---|---|
| | | Mean | Median | Mean | Median | Mean | Median |
| MONST3R (Zhang et al., 2025b) | ● | 0.851 | 0.689 | 1.734 | 0.958 | 1.292 | 0.823 |
| DUSt3R (Wang et al., 2024) | | 0.802 | 0.595 | 1.950 | 0.815 | 1.376 | 0.705 |
| CUT3R (Wang et al., 2025b) | | 0.458 | 0.342 | 1.633 | 0.792 | 1.042 | 0.567 |
| DAS3R Xu et al. (2024) | | 1.772 | 1.438 | 2.503 | 1.548 | **0.475** | **0.352** |
| VGGT (Wang et al., 2025a) | | 1.051 | 1.016 | 1.594 | 1.393 | 1.322 | 1.204 |
| **PAGE-4D(Ours)** | | **0.403** | **0.284** | **1.222** | **0.728** | 1.115 | 0.559 |

Completion (Comp.), and Overall error, where lower values indicate better reconstruction quality. As summarized in Tab. 3, our method achieves substantial improvements over prior feed-forward approaches. In particular, compared to the outputs produced by the point head of VGGT (Wang et al., 2025a), our approach reduces the mean Accuracy error by more than 60% (1.051 → 0.403) and the median error by over 70% (1.016 → 0.284). Similarly, our method yields consistent gains on Completion, with both mean and median errors reduced by over 20%. These results highlight the effectiveness of our dynamic-aware modeling: while existing methods either fail to accurately reconstruct moving regions (e.g., Easi3R (Chen et al., 2025)) or show degraded completion under dynamic motion (e.g., MonST3R (Zhang et al., 2025b)), our method balances accuracy and completeness, producing robust reconstructions in challenging dynamic scenarios.

## 4.5 DYNAMIC SCENES RENDERING

Rendering dynamic scenes has become a key focus in the computer vision community (Wu et al., 2024; Pumarola et al., 2021; Li et al., 2023; Zhou et al., 2023; 2024). However, most existing approaches rely heavily on accurate camera poses and high-quality initial point clouds—quantities that are often time-consuming to obtain and particularly challenging to estimate in the presence of complex object motion. PAGE-4D addresses this limitation by jointly predicting temporally consistent camera poses and dense 3D point clouds directly from RGB sequences containing dynamic content. To evaluate the utility of PAGE-4D for dynamic scene rendering, we use its reconstructed point clouds as initialization for the recent 4D-Gaussian splatting framework (Wu et al., 2024) and assess the resulting novel view synthesis quality on the Nerfie benchmark (Gafni et al., 2021). As shown in Tab. 4, our method consistently achieves superior rendering performance across scenes compared to prior feed-forward 3D reconstruction models. Notably, PAGE-4D provides a more robust geometric initialization which leads to improvements over both static-scene baselines (e.g., DUSt3R, VGGT) and recent dynamic-aware methods (e.g., MonST3R, CUT3R), demonstrating its effectiveness as a geometry prior for high-fidelity 4D rendering.

Table 4: **Novel View Synthesis on Nerfie (Gafni et al., 2021).** We report PSNR ↑, SSIM ↑, and LPIPS ↓ for each scene and the average.

| Method | chess4 | | | dvd | | | hand8 | | | laptop8 | | | tomato-mark8 | | | Avg | | |
|---|---|---|---|---|---|---|---|---|---|---|---|---|---|---|---|---|---|---|
| | PSNR ↑ | SSIM ↑ | LPIPS ↓ | PSNR ↑ | SSIM ↑ | LPIPS ↓ | PSNR ↑ | SSIM ↑ | LPIPS ↓ | PSNR ↑ | SSIM ↑ | LPIPS ↓ | PSNR ↑ | SSIM ↑ | LPIPS ↓ | PSNR ↑ | SSIM ↑ | LPIPS ↓ |
| dust3r | 11.572 | 0.263 | 0.633 | 12.363 | 0.494 | 0.546 | 12.445 | 0.269 | 0.639 | 11.818 | 0.185 | 0.622 | 14.961 | 0.361 | 0.564 | 12.632 | 0.314 | 0.601 |
| monst3r | 12.210 | 0.276 | 0.618 | 13.495 | 0.542 | 0.532 | 12.248 | 0.281 | 0.645 | 12.900 | 0.217 | 0.616 | 14.060 | 0.333 | 0.562 | 12.982 | 0.325 | 0.595 |
| cut3r | **17.480** | **0.448** | 0.525 | 14.856 | 0.528 | 0.465 | 14.466 | 0.365 | 0.559 | **16.505** | **0.437** | **0.365** | 17.289 | 0.461 | 0.447 | 16.319 | 0.448 | 0.472 |
| vggt | 16.807 | 0.390 | _0.497_ | _18.288_ | **0.676** | _0.379_ | _15.633_ | _0.510_ | _0.489_ | _15.954_ | _0.353_ | _0.475_ | _17.624_ | _0.488_ | _0.428_ | _16.861_ | _0.483_ | _0.454_ |
| **PAGE-4D(Ours)** | _17.338_ | _0.393_ | **0.491** | **18.355** | _0.671_ | **0.382** | **18.047** | **0.536** | **0.479** | 15.718 | 0.318 | 0.502 | **18.511** | **0.504** | **0.393** | **17.593** | **0.485** | **0.449** |

Table 5: **Video Depth Estimation on Sintel (Butler et al., 2012), Bonn (Palazzolo et al., 2019) and DyCheck (Yang et al., 2025).**

| Method | Align | Sintel | | Bonn | | DyCheck | |
|---|---|---|---|---|---|---|---|
| | | Abs Rel ↓ | $\delta < 1.25$ ↑ | Abs Rel ↓ | $\delta < 1.25$ ↑ | Abs Rel ↓ | $\delta < 1.25$ ↑ |
| VGGT* (Whole Model) | | _0.405_ | _0.593_ | 0.101 | _0.891_ | _0.175_ | _0.775_ |
| VGGT* (Middle Layers) | scale (Video-Depth) | 0.409 | 0.590 | _0.099_ | 0.879 | 0.177 | 0.766 |
| **Ours** - VGGT* (Middle Layers + Mask Attention) | | **0.357** | **0.699** | **0.092** | **0.904** | **0.170** | **0.785** |

## 4.6 ABLATION STUDIES

To evaluate the effectiveness of the proposed technique, we perform two ablation studies. First, we examine the fine-tuning strategy by comparing our approach—where only the middle $N_2$ attention layers are updated—with a baseline that fine-tunes all layers of the network (VGGT* (Whole Model)). Second, we study the role of the dynamic-aware aggregator by comparing our full model with a variant that simply fine-tunes the middle $N_2$ layers of VGGT without disentangling dynamics (VGGT* (Middle Layers)). From the comparison between VGGT* (Whole Model) and VGGT* (Middle Layers) as shown in Tab. 5, we observe that restricting fine-tuning to the middle layers yields performance comparable to full fine-tuning, confirming that these layers capture the most critical information. More importantly, by comparing **Ours** - VGGT* (Middle Layers + Mask Attention) with VGGT* (Middle Layers), we demonstrate that explicitly disentangling pose and geometry estimation through the dynamic-aware aggregator unlocks the potential of the backbone, leading to substantial performance gains.

## 5 CONCLUSION

Understanding dynamic scenes remains a central challenge in 4D computer vision, where object motion simultaneously provides valuable geometric cues and disrupts static-scene assumptions critical for camera pose estimation. In this work, we introduce PAGE-4D, a feedforward framework that adapts a pretrained 3D foundation model to dynamic environments through a disentanglement strategy. Our analysis shows that while VGGT excels in static scenarios, its unified treatment of motion leads to conflicts across tasks. To address this, we propose a dynamics-aware aggregator that disentangles static and dynamic content—suppressing dynamics for pose estimation while leveraging them for geometry and tracking. Combined with a targeted fine-tuning strategy on the most dynamic-sensitive layers, this design unlocks the backbone's latent capacity for handling motion. Extensive experiments demonstrate that PAGE-4D achieves state-of-the-art results across depth, pose, and point cloud reconstruction benchmarks. Importantly, we show that effective disentanglement enables strong generalization even with limited dynamic data, paving the way for scalable and efficient 4D scene understanding.

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
