APPENDIX

# 6 QUALITATIVE RESULTS

To further assess the generalization ability of PAGE-4D, we apply it to a diverse set of in-the-wild video sequences, as illustrated in Fig. 7. These examples span a wide range of dynamic scenarios, including human activities, object interactions, and complex background motions. We observe that PAGE-4D consistently produces stable and accurate predictions across all cases, effectively handling both static and highly dynamic regions. Notably, the method demonstrates strong robustness even under challenging conditions such as occlusions, fast motion, and varying illumination, highlighting its applicability beyond controlled benchmarks and into real-world video data.

# 7 ACKNOWLEDGMENT

All video materials used in this study were obtained from Pexels (https://www.pexels.com) and are distributed under the free Pexels License. While the license does not require individual attribution, we would like to acknowledge and thank the Pexels creators whose work contributed to the preparation of our figures and demonstrations.

# 8 ARCHITECTURE DESIGN

## 8.1 CHOICE OF MIDDLE-LAYER FINE-TUNING STRATEGY

To better understand the computational characteristics of our baseline, we analyze the parameter distribution of VGGT (Wang et al., 2025a), as summarized in Tab. 6. The majority of parameters are concentrated in the global attention blocks, which dominate both representational capacity and memory footprint. In contrast, the camera, depth, and point heads account for only a small fraction of the parameters, suggesting that most of the network capacity is devoted to learning general-purpose spatiotemporal features rather than task-specific decoding.

This observation leads to two key insights. First, fine-tuning the entire VGGT backbone is computationally inefficient: updating all attention layers substantially increases runtime and storage costs without yielding proportional gains. Second, since most parameters reside in global attention, selectively adapting the layers most sensitive to dynamic content is a more effective way to exploit the backbone's capacity.

Table 6: **Parameter distribution across modules.** "M" denotes millions of parameters.

| Module | Depth | Point | Track | Camera | Aggregator |
|---|---|---|---|---|---|
| **Parameters** | 32.7M | 32.7M | 65.9M | 216.2M | 909.1M |

To probe VGGT's handling of dynamic regions, we examine the role of attention maps within the global attention blocks—specifically at layers 4, 11, 17, and 23—which also provide inputs to the geometry decoder. To assess their contribution, we sequentially replace each layer's output with random noise and report the results in Tab. 7. Since only the last layer is fed into the camera head, randomizing other layers does not affect camera estimation performance; therefore, we omit camera estimation results here. The results show that the 17$^{th}$ layer exerts the strongest influence on geometry quality, underscoring the non-uniform importance of layers. Motivated by these findings, we propose a targeted fine-tuning strategy: adapting only

Table 7: **Study of Different Masking Strategies Applied to VGGT.** This experiment is conducted on the Odyssey dataset. We evaluate unscaled pose estimation using Relative Translation Error (RPE trans) and Relative Rotation Error (RPE rot). For static regions, Static-D denotes the Absolute Depth Error, and Static-T represents the Average Endpoint Error (EPE) for 2D point tracking.

| Method | RPE trans ↓ | RPE rot ↓ | Static-D ↓ | Static-T ↓ |
|---|---|---|---|---|
| Normal | 0.244 | 0.942 | 0.085 | 17.071 |
| Input-MSK | 0.246 | 1.006 | 0.099 | 17.866 |
| DD-MSK | 0.243 | 0.869 | 0.566 | 24.797 |
| w/o 4th | - | - | 0.114 | 18.821 |
| w/o 11th | - | - | 0.095 | 18.403 |
| w/o 17th | - | - | 1.663 | 39.841 |
| w/o 23rd | - | - | 0.103 | 17.645 |

Figure 6: **Visualization of learned masks.** Our dynamic mask prediction module effectively captures dynamic content in the scene without explicit supervision. (Best viewed in PDF.)

the middle 10 attention layers, which are most responsive to dynamic content. This achieves performance comparable to or better than full fine-tuning, while substantially reducing computational overhead.

During training, we observe that keeping the dynamic mask throughout the entire optimization process, or applying the mask only in the first stage and removing it in the second stage, yields similar performance. Notably, both strategies consistently outperform the baseline model trained with the original backbone without masking. Therefore, in our implementation, we provide both variants.

## 8.2 DESIGN OF POSE–GEOMETRY DISENTANGLEMENT

Figure 2(b) shows that dynamic objects consistently receive lower attention weights, indicating the model learns to suppress them rather than incorporate their motion. We therefore test baseline strategies inspired by prior work on motion-aware modeling Chen et al. (2025):

- Input-MSK (Input masking): Mask out dynamic regions at the image input level.
- DD-MSK (Dynamic–Dynamic masking): Suppress attention among dynamic tokens themselves.

Specifically, the DD-MSK strategy aims to mitigate the instability caused by excessive interactions within dynamic regions. Let $\mathcal{M}_{\text{Dynamic}} \in \mathbb{R}^{B \times (S \cdot P) \times C}$ denote the dynamic feature mask, where all patch tokens corresponding to dynamic regions are preserved. The DD-MSK is then constructed as:

$$\mathcal{M}_{\text{DD-MSK}} = \mathcal{M}_{\text{Dynamic}} \cdot \mathcal{M}_{\text{Dynamic}}^{\top}, \tag{7}$$

which blocks self-attention among dynamic patches while still allowing them to attend to static ones. In practice, this prevents dynamic regions from reinforcing noisy patterns, leading to more stable pose estimation and geometry reconstruction.

As shown in Table 7, DD-MSK improves pose estimation by isolating reliable motion cues, but simultaneously degrades geometry estimation, which requires integrating both static and dynamic motion signals. This observation aligns with our architectural design: dynamic information should be disentangled across tasks rather than globally suppressed.

**Summary.** Our structural and empirical analysis supports two conclusions: **1. Middle attention layers are more influential.** Perturbation analysis highlights the dominance of deeper global attention layers, particularly the 17[th], motivating fine-tuning only the middle 10 layers. **2. Rigid masking is suboptimal.** Suppressing dynamic patches improves pose estimation but harms geometry reconstruction, confirming the need for task-specific disentanglement.

## 9 TRAINING DETAILS

We train our model on a diverse mixture of dynamic and static datasets, including Odyssey, DynamicReplica, Kubric-MV, Spring, CO3D, Waymo, Sintel and our internal dataset, with a total of approximately 2.39M sampled sequences as in Tab. 8. To balance domain diversity, we assign per-dataset sampling multipliers and cap the number of training batches per epoch. Images are resized to $518 \times 518$ with patch size 14. We adopt AdamW with an initial learning rate of $1 \times 10^{-5}$, weight decay 0.01, and gradient clipping of 1.0. Mixed precision (bfloat16) training is used to improve efficiency. Following our middle-layer fine-tuning strategy, we freeze the shallow and final blocks of VGGT while updating only the middle 10 global attention layers, which we identify as most sensitive to dynamic content. The multitask loss combines camera, depth, and point supervision with weights of 5.0, 1.0, and 1.0, respectively.

## 10 ARCHITECTURE ANALYSIS

To further evaluate the effectiveness of our dynamic mask prediction module, we visualize the predicted masks on sequences from the Odyssey dataset. As shown in Fig. 6, the learned masks successfully highlight moving objects such as people and vehicles, while leaving static backgrounds largely unmarked. Importantly, this separation emerges without any explicit supervision, indicating that the model is able to infer dynamic regions purely from motion cues and spatiotemporal inconsistencies. This validates our design choice: the dynamic mask prediction module provides a reliable mechanism to disentangle dynamic and static content, thereby improving the robustness of pose estimation and geometry reconstruction in challenging dynamic scenes.

Table 8: **Datasets used in the fine-tuning process. Dynamic** indicates whether the dataset contains dynamic objects. **Frames** and **Scenes** denote the number of image frames and unique object-centric scenes. **Ratio** is the scene-level sampling multiplier used to balance datasets during training.

| Dataset | Dynamic | Realistic | Frames | Scenes | Ratio |
|---|---|---|---|---|---|
| CO3D Reizenstein et al. (2021) | × | ✓ | 1.5M | 19K | 20% |
| PointOdyssey Zheng et al. (2023) | ✓ | × | 6K | 131 | 10% |
| Kubric-MV Greff et al. (2022) | ✓ | × | 70K | 3K | 10% |
| DynamicReplica Karaev et al. (2023) | ✓ | × | 145K | 484 | 20% |
| Spring Mehl et al. (2023) | ✓ | × | 200K | 37 | 10% |
| Waymo Sun et al. (2020) | ✓ | ✓ | 230K | 1.1K | 10% |
| Ours | ✓ | ✓ | 480K | 2K | 20% |

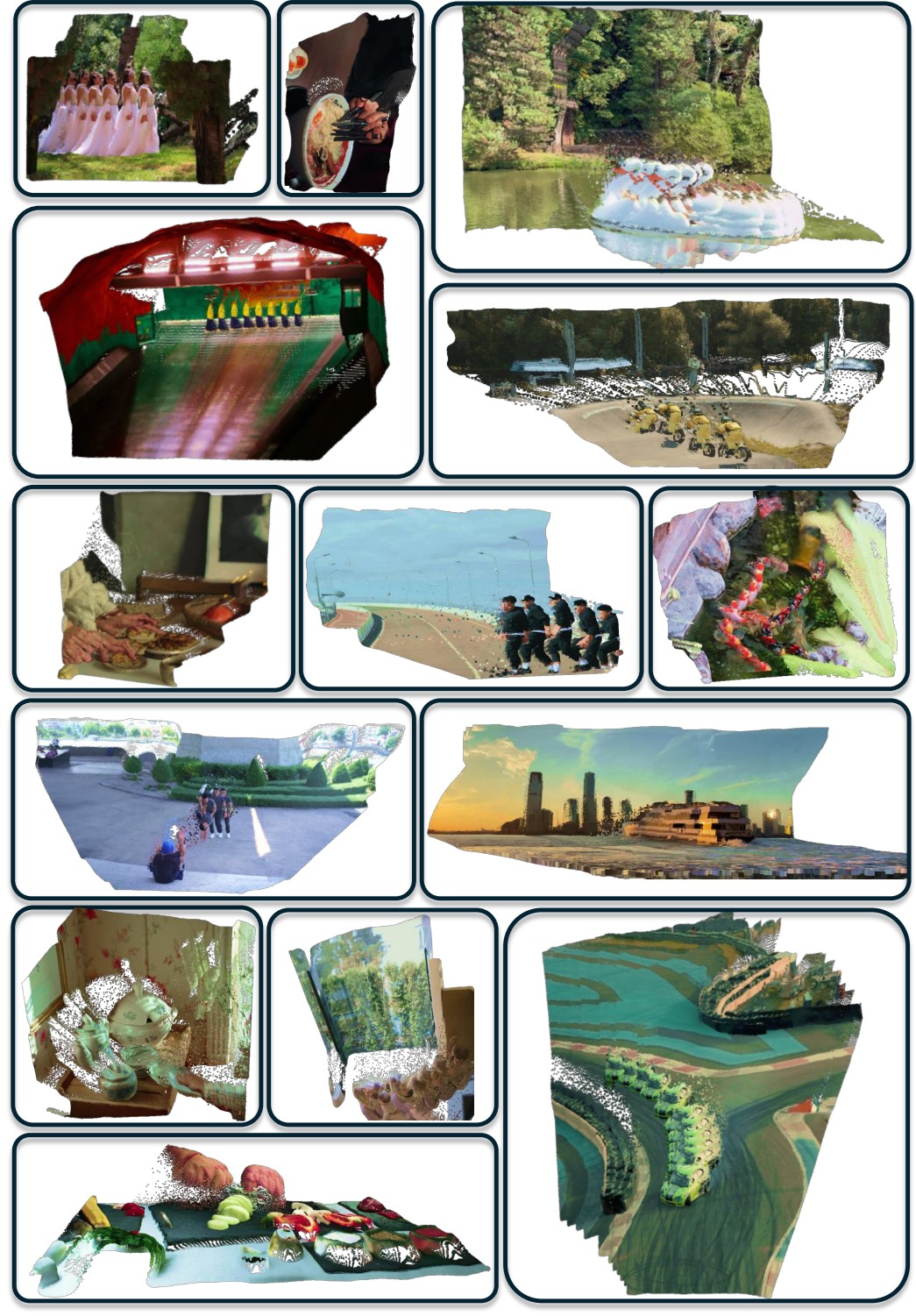

Figure 7: **PAGE-4D** takes a sequence of RGB images depicting a dynamic scene as input and simultaneously predicts the corresponding camera parameters and 3D geometry information—all within a fraction of a second. (Best viewed in PDF.)