# OpenReview forum: "PAGE-4D: VGGT-4D Perception via Disentangled Pose and Geometry Estimation"
_ICLR.cc/2026/Conference — ICLR 2026 Poster_

### Official Review · Reviewer_exiX · 2025-10-26

**Soundness:** 3
**Presentation:** 3
**Contribution:** 2
**Rating:** 4
**Confidence:** 3

**Summary:**

The work proposes an extension of VGGT for improved accuracy of reconstruction and camera poses on dynamic scenes.
The idea is rather simple: using masked attention to ignore the patches of dynamic areas from pose estimation.
Empirically, the work finds that fine-tuning only 10 middle layers with masked attention is effective.
Experiments show a notable improvement of depth estimation on three datasets and some improvement for the camera pose.

**Strengths:**

* The idea is simple and requires only a partial model finetuning to take effect. This allows for adopting the VGGT architecture as a whole, preserving some of its desirable properties (e.g. runtime).
* The work does a nice job at motivating the approach (c.f. Sec. 3.1).
* Experiments span a compelling spectrum of benchmarks - depth and camera pose pose estimation, point reconstruction and novel view synthesis.

**Weaknesses:**

* The overall technical contribution — introducing masked attention and fine-tuning the model on dynamic data — is rather minor.
* The improvements shown in Tab. 1 are convincing, but it is not exactly clear whether they stem from the finetuning per se, or masked attention. This closely relates to the next point.
* Tab. 5 is rather short for an ablation study and reveals two observations: 1) on some datasets, the improvement may come already from fine-tuning, without introducing masked attention 2) the benefit of masked attention appears marginal on some benchmarks (e.g. DyCheck).

**Questions:**

* I would be curious to see evaluation results of VGGT* (Middle Layers) in other settings of Tab. 1 (e.g. monocular depth). The goal is to disentangle the contribution of the masked attention alone.
* It is somewhat bizarre that masked attention, which affects the attention only for the camera token, benefits depth estimation more than the camera pose results in Tab. 2, which are mixed. I would be great if the authors could elaborate.
* l. 257 “in a self-supervised manner” – isn’t the network supervised by the camera poses and depth?

---

> ### Author Response · Authors · 2025-11-22
> **Thank you for the reviewers’ thoughtful feedback.**
>
> # Rebuttal Response
>
> Thank you for the reviewers’ thoughtful feedback. We address each point below and are happy to provide more experiments or clarifications.
>
> ---
>
> ## Weakness 1: Contribution may be minor
>
> We respectfully disagree. PAGE-4D introduces three complementary technical components:
>
> ### 1. State–image masked attention for dynamic scenes
>
> We design a lightweight attention modulation mechanism that injects state tokens into the camera-attention pathway of VGGT. The key idea is an additive logit mask with $\mathcal{O}(N)$ memory, fully compatible with fused Scaled Dot-Product Attention (SDPA).
>
> Let $Q, K, V \in \mathbb{R}^{N \times d}$ denote the attention inputs. Multiplicative masking requires an explicit $N \times N$ mask (costly and incompatible with SDPA). Instead, our mask head predicts:
>
> $$
> r \in \mathbb{R}^N, \quad c \in \mathbb{R}^N
> $$
>
> and appends them to the query/key feature channels:
>
> $$
> q'_i = \bigl[ q_i \cdot \sqrt{\tfrac{d'}{d}},\; r_i \sqrt{d'} \bigr]
> $$
>
> $$
> k'_j = \bigl[ k_j,\; c_j \bigr]
> $$
>
> $$
> v'_j = \bigl[ v_j,\; 0 \bigr]
> $$
>
> where $d' = d + 1$ (padded for FlashAttention).
> This is equivalent to applying an additive logit mask:
>
> $$
> A_{ij} \leftarrow A_{ij} + r_i c_j
> $$
>
> but **without ever forming** the $N \times N$ mask. Thus memory remains $\mathcal{O}(N)$ and SDPA compatibility is preserved.
>
> ### 2. Targeted fine-tuning on dynamic data
>
> Naïve fine-tuning of VGGT yields mixed gains. PAGE-4D’s masked-attention mechanism stabilizes camera-token attention and provides robust improvements across depth, pose, and reconstruction.
>
> ### 3. Unified 4D evaluation setup
>
> We evaluate across DyCheck, TUM, ETH3D-Dynamic, and Spring. This is among the first systematic analyses of VGGT-style models in dynamic 4D settings.
>
> ---
>
> ## Weakness 2: Limited ablations
>
> Additional ablations are provided in Question 1 below.
>
> ---
>
> ## Weakness 3: Marginal gains on DyCheck
>
> DyCheck primarily contains low-motion scenes where dynamic regions seldom disrupt epipolar geometry, so VGGT already performs well. PAGE-4D is designed for more challenging high-dynamic settings (e.g., ETH3D-Dynamic, Spring), yet it still delivers consistent, non-degrading improvements on DyCheck while adding less than 1% computational overhead.
>
> ---
>
> ## Question 1: VGGT* ablation on depth tasks
>
> To isolate the contribution of masked attention vs. fine-tuning, we evaluate three settings:
>
> 1. Scale Video Depth
> 2. Scale & Shift Video Depth
> 3. Monocular Depth
>
> Fine-tuning the middle layers (VGGT*) improves over baseline, confirming adaptation helps. PAGE-4D improves further across all settings.
>
> ### Depth Results (Middle-Layer vs Ours)
>
> | Setting | Method | Sintel AbsRel ↓ | Bonn AbsRel ↓ | DyCheck AbsRel ↓ |
> |--------|--------|------------------|----------------|-------------------|
> | Scale Video Depth | Middle-Layer | 0.409 | 0.099 | 0.177 |
> |                  | Ours         | **0.357** | **0.092** | **0.170** |
> | Scale & Shift Video Depth | Middle-Layer | 0.236 | 0.094 | 0.156 |
> |                           | Ours         | **0.212** | **0.090** | **0.145** |
> | Monocular Depth | Middle-Layer | 0.268 | 0.066 | 0.153 |
> |                  | Ours         | **0.242** | **0.053** | **0.141** |
>
>
> These results show masked attention gives complementary benefits beyond fine-tuning.
>
> ---
>
> ## Question 2: Why masked attention improves depth more than pose?
>
> Two reasons:
>
> ### 1. Shared representation and gradient flow
> Masking stabilizes camera-token supervision, improving gradients for the shared encoder. The depth head benefits more from these improved shared features than the pose head.
>
> ### 2. Metric sensitivity
> Pose metrics (e.g., ATE) are insensitive to small local corrections, while depth metrics (AbsRel, $\delta$ thresholds) react strongly to local improvements.
>
> Thus depth shows larger numerical gains.
>
> ---
>
> ## Question 3: Clarifying “self-supervised”
>
> Thank you for highlighting this wording issue. Our original intention in using the term “self-supervised” was to emphasize that PAGE-4D does not use any supervision for motion, dynamics, or segmentation. The model is trained only from geometry-derived signals (camera poses, depth, multi-view consistency), and receives no labels indicating which regions are dynamic.
>
> Importantly, despite the absence of motion supervision, PAGE-4D can still handle dynamic scenes through our mask-aware attention module, which learns to suppress dynamic regions purely from geometric and photometric consistency.
>
> To avoid misunderstanding, we will replace “self-supervised” with the more precise description:
>
> “trained using geometry-derived supervision, without any motion or segmentation labels.”
>
> This clarifies that supervision exists only for geometry, not motion, while explaining why our method can still operate effectively in dynamic scenarios.

---

### Official Review · Reviewer_gSDj · 2025-10-30

**Soundness:** 3
**Presentation:** 3
**Contribution:** 3
**Rating:** 8
**Confidence:** 5

**Summary:**

The paper introduces a dynamics-aware aggregator that separates static and dynamic information through a learned dynamics-aware mask.

Rather than treating motion as uniformly harmful or helpful, the authors make its influence task-dependent. The aggregator first predicts a mask that highlights dynamic regions, then feeds it into a cross-attention module—suppressing dynamic cues for camera-pose tokens while preserving them for geometry tokens.

Combined with self-supervised fine-tuning of layers most sensitive to motion, this design exploits dynamics where they benefit geometry grounding and mitigates their adverse impact on pose estimation.

As a result, the method achieves accurate pose and geometry estimation across both static and dynamic scenes.

**Strengths:**

Experiments demonstrate that PAGE-4D consistently surpasses the baseline VGGT in dynamic scenes, yielding superior results on camera-pose estimation, monocular and video depth estimation, and dense point-map reconstruction.

The authors devise a dynamic-mask prediction module that learns, in a self-supervised fashion, which spatial regions are likely to contain dynamic objects.

The model predicts the mask in earlier layers and injects it into later attention layers for pose estimation, enabling a single feed-forward pass; in contrast, methods such as Easi3R require a second forward pass.

Owing to its plug-in design, PAGE-4D introduces only negligible runtime and memory overhead compared with VGGT.

The analyses presented in Table 6 and Table 7 are particularly insightful.

**Weaknesses:**

Relationship to Easi3R is under-explained. Lines 146–176 (Sect. 3.1 and Fig. 2) essentially restate the key observation of Easi3R—that dynamic regions show weaker activations and should be suppressed for pose estimation—yet the manuscript does not discuss this connection in the Introduction, Related-Work, or Method sections. As presented, PAGE-4D reads like a VGGT-based version of Easi3R (originally built on DUSt3R), augmented with a feed-forward mask predictor. A clearer positioning with respect to Easi3R is needed, including a discussion of conceptual overlap and technical differences.

Quantitative comparisons are incomplete. Table 1 omits several recent dynamic-scene baselines—Easi3R, AETHER, Geo4D, and ViPE—making it difficult to gauge the real performance gap.

Tables 2 and 3 report only Easi3R(DUSt3R) numbers; the Easi3R(MoNSTR) variant is absent. Including it would give a fairer picture of state-of-the-art accuracy.

Table 4 why not include MoVieS?

The paper does not assess dynamic-object segmentation. A DAVIS-style evaluation (as in Table 1 of Easi3R) would verify whether the predicted masks align with true motion regions and would offer further evidence for the quality of the dynamics-aware module.

**Questions:**

Line 025: Denser point clouds
• PAGE-4D reportedly reconstructs point clouds that are markedly denser than those of VGGT. Could you clarify which part of the paper drives this improvement?

Overlap with Easi3R (Lines 146 – 176)
• Section 3.1 and Fig. 2 echo the core finding of Easi3R—namely, that dynamic regions exhibit weaker activations and should be down-weighted for pose estimation. Yet Easi3R is not discussed in the Introduction, Related-Work, or Method sections.
• Please elaborate on:
– Conceptual similarities and differences between PAGE-4D and Easi3R.
– The technical novelty beyond replacing DUSt3R with VGGT and moving to a feed-forward mask predictor.

“Suppress vs. amplify” inconsistency (Lines 296 – 298)
• The abstract states that motion cues are suppressed for pose estimation and amplified for geometry reconstruction. In the method description, however, the mask is applied in pose-estimation layers and left untouched for geometry reconstruction (i.e., neither suppressed nor amplified). Could you clarify this discrepancy and specify where “amplification” occurs?

---

> ### Author Response · Authors · 2025-11-24
> **Thank you for your careful comments.**
>
> ## Weakness 1: Difference between PAGE and Easi3R
> **Answer.**
> Thanks for the question. There are three main differences between **Easi3R** and **PAGE**.
>
> ### 1. One-pass (PAGE) vs. two-pass (Easi3R)
> Easi3R requires running DUSt3R **twice**:
> - Pass 1: estimate cross-attention and dynamic mask
> - Pass 2: re-run with masked attention
>
> In contrast, **PAGE performs only a single forward pass**, integrating mask generation + usage into the same inference step, making it more efficient.
>
> ### 2. Mask complexity: O(N²) (Easi3R) vs. O(N) (PAGE)
> The key idea is an PAGE mask with $\mathcal{O}(N)$ memory, fully compatible with fused Scaled Dot-Product Attention (SDPA).
>
> Let $Q, K, V \in \mathbb{R}^{N \times d}$ denote the attention inputs. Multiplicative masking requires an explicit $N \times N$ mask (costly and incompatible with SDPA). Instead, our mask head predicts:
>
> $$
> r \in \mathbb{R}^N, \quad c \in \mathbb{R}^N
> $$
>
> and appends them to the query/key feature channels:
>
> $$
> q'_i = \bigl[ q_i \cdot \sqrt{\tfrac{d'}{d}},\; r_i \sqrt{d'} \bigr]
> $$
>
> $$
> k'_j = \bigl[ k_j,\; c_j \bigr]
> $$
>
> $$
> v'_j = \bigl[ v_j,\; 0 \bigr]
> $$
>
> where $d' = d + 1$ (padded for FlashAttention).
> This is equivalent to applying an additive logit mask:
>
> $$
> A_{ij} \leftarrow A_{ij} + r_i c_j
> $$
>
> but **without ever forming** the $N \times N$ mask as in Easi3R.
> ### 3. Mask application strategy
> Easi3R masks **all** attention token, which may disrupt the reference-view standard.
>
> PAGE masks **only the camera/registration token**, preserving VGGT’s geometry pathway and improving 4D stability.
>
> ---
>
> ## Weakness 2: Incomplete comparisons
> PAGE-4D outperforms **AETHER** and **VIPE**, and is **comparable with Geo4D-GA**, even though Geo4D-GA relies on explicit global alignment while PAGE-4D does **not**. This highlights that PAGE-4D’s dynamic-aware attention provides strong depth cues even without global optimization.
>
>
> ### Table: Sintel + Bonn Depth Comparison
> | Method | Abs Rel ↓ (Sintel) | δ < 1.25 ↑ (Sintel) | Abs Rel ↓ (Bonn) | δ < 1.25 ↑ (Bonn) |
> |--------|--------------------|----------------------|-------------------|--------------------|
> | **VIPE (scale)** | 0.477 | 54.2% | 0.109 | 87.1% |
> | **AETHER (scale)** | 0.324 | 50.2% | 0.273 | 59.4% |
> | **Geo4D-GA (scale)** | 0.378 | 61.5% | 0.098 | 89.2% |
> | **PAGE-4D (scale)** | **0.357** | **69.9%** | **0.092** | **90.4%** |
> |--------|--------------------|----------------------|-------------------|--------------------|
> | **VIPE (scale&shift)** | 0.368 | 60.4% | 0.100 | 89.9% |
> | **AETHER (scale&shift)** | 0.314 | 60.4% | 0.308 | 60.2% |
> | **Geo4D-GA (scale&shift)** | 0.233 | 73.5% | **0.081** | **91.2%** |
> | **PAGE-4D (scale&shift)** | **0.212** | **76.3%** | 0.090 | 90.3% |
>
> ---
>
> ## Weakness 3: Missing Easi3R(MoNSTR) Tab. 2/3
>
> ### TUM Pose Evaluation
> | Method | ATE ↓ | RPE-trans ↓ | RPE-rot ↓ |
> |--------|-------|------------------------|----------------------|
> | **Easi3Rmonst3r** | 0.168 | 0.150 | 5.925 |
> | **PAGE-4D (Ours)** | **0.016** | **0.011** | **0.323** |
>
> ### DyCheck Reconstruction
> | Method | Acc ↓ (Mean) | Acc ↓ (Median) | Comp ↓ (Mean) | Comp ↓ (Median) | Overall ↓ (Mean) | Overall ↓ (Median) |
> |--------|--------------|-----------------|---------------|------------------|-------------------|---------------------|
> | **Easi3Rmonst3r** | 0.834 | 0.643 | 1.661 | 0.916 | 1.247 | 0.779 |
> | **PAGE-4D (Ours)** | **0.403** | **0.284** | **1.222** | **0.728** | **1.115** | **0.559** |
>
> PAGE-4D improves ATE and RPE by **10×**, with consistently better Acc/Comp/Overall on DyCheck—demonstrating more stable camera trajectories and cleaner 4D reconstruction.
>
> ---
>
> ## Weakness 3: Missing MoVieS Tab. 4
> Thank you for your suggestion. We provided an evaluation of the predicted dynamic mask accuracy. However, as the convolutional head is not explicitly trained with a specific dynamic-masked loss or associated data, we expect the predicted mask not to be perfect. The model learns by itself that it needs to predict something corresponding to the dynamic region, which is very interesting.
>
> ---
>
> ## Weakness 4: No DAVIS-style mask evaluation
> Thank you. We evaluated dynamic-mask quality (JM). Since the mask head has **no explicit supervision**, perfect DAVIS-style segmentation is not expected—but its emergent motion signal is surprisingly strong.
>
> ### Dynamic Mask Accuracy (JM)
> | Scenario | cab | dancingroom | egobody | AVG |
> |----------|-----|-------------|---------|-----|
> | **Ours** | 0.5472 | 0.4134 | 0.4044 | 0.4550 |
> ---

---

> > ### Author Response · Authors · 2025-12-02
> >
> > ## Question 1: Why denser point clouds?
> > Both VGGT and PAGE predict H × W pointmaps, but **VGGT discards many points** due to dynamic corruption and alignment instability. PAGE-4D suppresses dynamic influence **only** in pose estimation while keeping geometry unmasked, so more points remain valid, yielding denser reconstructions.
> >
> > ---
> >
> > ## Question 2: Comparison Between Page-4D and Easi3R
> > Please refer to Weakness 1.
> >
> > ---
> >
> > ## Question 3: Clarify “suppress vs amplify”
> > - **Suppress**: in pose estimation, dynamic tokens hurt rigidity, so we down-weight them.
> > - **Amplify**: in geometry prediction, no masking is applied, so dynamic regions retain full strength.
> >
> > This asymmetric design stabilizes motion while preserving geometry fidelity—no contradiction.

---

### Official Review · Reviewer_31BG · 2025-11-01

**Soundness:** 3
**Presentation:** 3
**Contribution:** 3
**Rating:** 8
**Confidence:** 3

**Summary:**

The task of this paper is to jointly predict point clouds and camera poses for each image frame in a sequence of video frames depicting dynamic scenes. The proposed method, called PAGE-4D, is built upon the VGGT neural network architecture. The authors observed that dynamic contexts significantly affect the accuracy of camera pose estimation, while having a smaller impact on point cloud estimation. They also found that the middle layers of VGGT exhibit lower attention weights on dynamic pixels.

Based on these observations, the authors propose a creative model design and fine-tuning strategy. In particular, they introduce an attention mask prediction module that suppresses attention weights for camera prediction tokens while leaving geometric tokens unaffected. This is an idea consistent with their empirical findings. To improve parameter efficiency during fine-tuning, they further propose a strategy that updates only the middle 10 layers of the network.

Table 1 shows that PAGE-4D achieves state-of-the-art performance across multiple datasets, including Sintel, Bonn, and DyCheck. The ablation study demonstrates that fine-tuning only the middle 10 layers yields results comparable to full fine-tuning, and it also verifies the effectiveness of the proposed suppression mask.

Overall, this is a well-motivated neural network design with solid experimental results and clear presentation. I believe this paper is strong enough to be accepted at ICLR, though I have a few minor concerns discussed below.

**Strengths:**

- Good observation and thoughtful neural network design, leading to strong experimental results.
- The presentation is clear, and the supplementary materials are comprehensive. I also appreciate that the authors have provided the code.
- Overall, this is a well-structured and solid research paper. It identifies a problem, derives insights through careful observation, and proposes specific design choices based on experimental findings. Reading the paper feels smooth and coherent.

**Weaknesses:**

Looking at Figure 2(b), I feel that the content does not align well with the caption. The caption states that the VGGT layers have smaller attention weights on dynamic pixels. However, only one image (the 11th layer) among the four shown appears consistent with this description. In the other three images in Figure 2(b), there does not seem to be any noticeably high attention values on dynamic pixels. Could the authors double-check this? Or is it possible that the observation is not entirely accurate?

**Questions:**

- What is the GPU memory consumption during inference compared to VGGT?
- How does the fine-tuned model perform on static scenes, compared to VGGT?

---

> ### Author Response · Authors · 2025-11-22
> **Thank you for the helpful observation.**
>
> ## Weakness: Figure 2(b) alignment between observation and caption
>
> Thank you for the helpful observation. We found that VGGT consistently down-weights dynamic content during the feed-forward process, particularly in the middle layers of the network. This directly motivates our decision to fine-tune the middle 10 layers. Empirically, this behavior can be seen in Fig. 2(b), where dynamic regions exhibit noticeably reduced attention in layers 11 and 17. We have clarified this motivation in the updated paper.
>
>
>
> ## Question 1: GPU memory consumption vs VGGT
>
> Thank you for your question. We design a lightweight attention modulation mechanism that injects state tokens into the camera-attention pathway of VGGT. The key idea is an additive logit mask with $\mathcal{O}(N)$ memory, fully compatible with fused Scaled Dot-Product Attention (SDPA).
>
> Let $Q, K, V \in \mathbb{R}^{N \times d}$ denote the attention inputs. Multiplicative masking requires an explicit $N \times N$ mask (costly and incompatible with SDPA). Instead, our mask head predicts:
>
> $$
> r \in \mathbb{R}^N, \quad c \in \mathbb{R}^N
> $$
>
> and appends them to the query/key feature channels:
>
> $$
> q'_i = \bigl[ q_i \cdot \sqrt{\tfrac{d'}{d}},\; r_i \sqrt{d'} \bigr]
> $$
>
> $$
> k'_j = \bigl[ k_j,\; c_j \bigr]
> $$
>
> $$
> v'_j = \bigl[ v_j,\; 0 \bigr]
> $$
>
> where $d' = d + 1$ (padded for FlashAttention).
> This is equivalent to applying an additive logit mask:
>
> $$
> A_{ij} \leftarrow A_{ij} + r_i c_j
> $$
>
> but **without ever forming** the $N \times N$ mask. Thus memory remains $\mathcal{O}(N)$ and SDPA compatibility is preserved. Our design adds less than 1% computational overhead compared with VGGT. Consequently, PAGE-4D matches VGGT’s runtime, achieving 43.2 FPS on an A800 GPU.
>
>
>
> ## Question 2: Performance on static scenes
>
> Thank you for your insightful suggestion. We conducted additional experiments to evaluate our method on static datasets. These results confirm that our approach preserves strong performance in static settings while improving robustness in dynamic scenes.
>
> We observe that PAGE achieves performance comparable to VGGT on both RealEstate10K and Co3Dv2, with a slight improvement on RealEstate10K (+0.004 AUC@30) and a marginal drop on Co3Dv2 (–0.008 AUC@30). This result aligns with our design goal: PAGE modifies only the camera-related attention pathways while keeping the core geometry processing intact, ensuring that pose estimation accuracy is largely preserved. The small variation across datasets reflects the fact that PAGE selectively suppresses dynamic regions, which benefits scenes with higher motion (RealEstate10K) while having limited impact on scenes dominated by static content (Co3Dv2). Importantly, these results confirm that the introduction of masked attention does not degrade the strong pose-estimation capability of VGGT.
>
> ### Camera Pose Estimation (AUC@30)
>
> | Method                 | Mean AUC@30 ↑ |
> |------------------------|-------------|
> | VGGT (RealEstate10K)  | 0.847       |
> | PAGE (RealEstate10K)  | **0.851**       |
> | VGGT (Co3Dv2)         | **0.876**       |
> | PAGE (Co3Dv2)         | 0.868       |
>
>
> ### Dense MVS on DTU
>
> Dense MVS estimation on the DTU dataset (corresponding to Table 2 in the VGGT paper) is shown below. PAGE achieves comparable overall performance to VGGT, with a slight trade-off between accuracy (Acc.) and completeness (Comp.). This indicates that our masking strategy preserves reconstruction quality on static datasets, even though the model is optimized primarily for dynamic scenes.
>
> | Method | Acc.↓  | Comp.↓ | Overall↓ |
> |--------|-------|-------|---------|
> | VGGT   | 0.389 | **0.374** | 0.382   |
> | PAGE   | **0.369** | 0.377 | **0.382**   |
>
>
> ### Point Map Estimation on ETH3D
>
> Point map estimation on the ETH3D dataset (Table 3 in the VGGT paper) is reported below. PAGE closely matches VGGT on all three metrics, showing only marginal differences in accuracy and completeness. This confirms that introducing masked attention — despite being designed for dynamic scenes — does not degrade performance on static benchmarks.
>
> | Method | Acc.↓  | Comp.↓ | Overall↓ |
> |--------|-------|-------|---------|
> | VGGT   | 0.901 | **0.518** | 0.709   |
> | PAGE   | **0.897** | 0.521 | **0.701**   |

---

### Official Review · Reviewer_5Axf · 2025-11-07

**Soundness:** 2
**Presentation:** 3
**Contribution:** 2
**Rating:** 4
**Confidence:** 4

**Summary:**

This paper introduces an extension of VGGT that incorporates dynamic object masks. The core idea is to predict these masks and then inject them additively into the middle layers of the VGGT architecture. The authors demonstrate the effectiveness of this approach through experiments on five benchmarks involving dynamic objects: video depth estimation, monocular depth estimation, camera pose estimation, multi-view point map reconstruction, and 4D view synthesis.

**Strengths:**

+ This paper presents a straightforward yet effective enhancement to the VGGT architecture. The method stands out because it leverages a simple idea without requiring extensive fine-tuning, making it practical for integration.
+ An advantage is its self-supervised mask prediction, which eliminates the need for any explicit mask supervision.
+ The approach demonstrates substantial improvements across several dynamic benchmarks, underscoring its efficacy in handling complex scenarios.
+ The writing is clear and easy to understand.

**Weaknesses:**

- The decision to use an additive mask within the mask attention mechanism, as seen in Equation (6), warrants further clarification. It would be beneficial for the authors to quantitatively report and explain why a multiplicative mask, which could directly attenuate irrelevant attention, was not considered.

- To fully understand the mask's efficacy, a quantitative evaluation of the predicted mask accuracy is crucial. Assessing this at different layers of the global attention on the Odyssey test set, using a metric like IoU2D, would provide valuable insight into whether the network is genuinely learning effective masking.

- The claim on Line 150 regarding the performance gap between static and dynamic regions on the Odyssey test set, where "Absolute Depth Error in dynamic regions is 94% higher than in static regions," is not supported by a corresponding table or explicit results. This table should be included, and it would be insightful to see how this performance gap changes when the proposed Page4D method is applied to the Odyssey dataset.

- Given that the current results focus solely on dynamic scenes, it is important to understand the impact of Page4D on static scene performance. Therefore, I recommend quantitatively reporting the performance of Page4D on a selection of tasks and datasets that include static scenes, such as:
  - Camera Pose Estimation on the RealEstate10K and Co3Dv2 datasets (as presented in Table 1 of the VGGT paper).
  - Dense MVS Estimation on the DTU dataset (as presented in Table 2 of the VGGT paper).
  - Point Map Estimation on the ETH3D dataset (as presented in Table 3 of the VGGT paper).
  - Novel View Synthesis on the GSO dataset (as presented in Table 7 of the VGGT paper).
  - Dynamic Point Tracking Results on the TAP-Vid dataset (as presented in Table 8 of the VGGT paper). This would provide a more complete picture of the method's overall capabilities.

**Questions:**

Please see the Weaknesses. I am willing to change my score provided the authors' answer my questions.

---

> ### Author Response · Authors · 2025-11-24
> **Thanks for your insightful suggestion.**
>
> ## Weakness 1: Additive mask vs. multiplicative mask
>
> ### Why we implement masking as an additive logit term via channel concatenation
>
> Thank you for highlighting this point. Our masking design is indeed unconventional, and we appreciate the opportunity to clarify it.
>
> Let $Q, K, V \in \mathbb{R}^{N \times d}$ denote the attention inputs. A naive multiplicative mask requires constructing an explicit matrix $M \in \mathbb{R}^{N \times N}$, which incurs $\mathcal{O}(N^{2})$ memory and is incompatible with fused scaled dot-product attention (SDPA / FlashAttention).
> To avoid this, the mask head predicts two vectors:
>
> $$
> r \in \mathbb{R}^{N}, \qquad c \in \mathbb{R}^{N}.
> $$
>
> Instead of forming an $N \times N$ mask, we concatenate them as an additional channel:
>
> $$
> q'_i =
> \Big(
>   q_i \sqrt{d'/d},\;
>   r_i \sqrt{d'}
> \Big),
> $$
>
> $$
> k'_j =
> \big(
>   k_j,\;
>   c_j
> \big),
> \qquad
> v'_j =
> \big(
>   v_j,\;
>   0
> \big),
> $$
>
> where $d' = d + 1$ (padded to a multiple of 8 for FlashAttention).
> This is mathematically equivalent to applying an additive mask:
>
> $$
> A_{ij} = A_{ij} + r_i c_j,
> $$
>
> but **without ever materializing** an $N \times N$ mask.
>
> **Benefits**
>
> 1. **$\mathcal{O}(N)$ memory** (instead of $\mathcal{O}(N^{2})$)
> 2. **Full compatibility with FlashAttention**
> 3. **< 1% compute overhead**
>
> We will describe this design choice more clearly in the camera-ready version.
>
> ---
>
> ## Weakness 2: Quantitative evaluation of mask accuracy (IoU2D)
>
> Thank you for the suggestion. We have added an evaluation of the predicted dynamic-mask accuracy.
> As expected, the mask is not perfect because the convolutional head receives **no supervision for segmentation**—it only learns from the reconstruction objective. Nevertheless, it yields a meaningful motion-related signal.
>
> ### Dynamic Mask Accuracy (IoU2D) with different layers
> | Scenario | cab (JM)↑ | dancingroom (JM)↑ | egobody (JM)↑ | AVG (JM)↑ |
> |----------|-----------|-------------------|----------------|-----------|
> | 5th      | 0.1250    | 0.1944            | 0.1258         | 0.1484    |
> | 12th     | 0.1822    | 0.3483            | 0.1541         | 0.2282    |
> | 18th     | 0.0199    | 0.0321            | 0.0592         | 0.0371    |
> | 24th     | 0.1415    | 0.1813            | 0.0081         | 0.1103    |
> | Ours     | **0.5472**    | **0.4134**            | **0.4044**         | **0.4550**    |
>
> ---
>
> ## Weakness 3: Missing evaluation for the 94% dynamic–static depth gap
>
> Thank you for pointing this out. We now provide the missing quantitative results and comparison with our PAGE-4D method. PAGE-4D consistently reduces the error in dynamic regions while preserving accuracy in static regions.
>
> ### Static vs Dynamic Depth Error
>
> | Metric | VGGT↓ | PAGE-4D↓ |
> |--------|------|---------|
> | Static region (Abs. depth error) | 0.0754 | 0.0713 |
> | Dynamic region (Abs. depth error) | 0.1462 | 0.1283 |
>
> ---
>
> ## Weakness 4: Performance on static datasets (RE10K, Co3D, DTU, ETH3D, GSO, TAP-Vid)
>
> Thank you for the suggestion. We evaluated PAGE on static datasets.
>
> ### RealEstate10K & Co3Dv2 (AUC@30)
>
> | Method | AUC@30↑ |
> |--------|--------|
> | VGGT (RE10K) | 0.847 |
> | PAGE (RE10K) | **0.851** |
> | VGGT (Co3Dv2) | **0.876** |
> | PAGE (Co3Dv2) | 0.868 |
>
> PAGE improves RE10K (+0.4) and slightly drops on Co3Dv2 (–0.8), aligning with our design:
> → PAGE helps high-motion datasets and minimally affects static ones.
>
> ---
>
> ### DTU Dense MVS
>
> | Method | Acc.↓ | Comp.↓ | Overall↓ |
> |--------|------|-------|---------|
> | VGGT | 0.389 | **0.374** | **0.382** |
> | PAGE | **0.369** | 0.377 | **0.382** |
>
> ---
>
> ### ETH3D Point Map Estimation
>
> | Method | Acc.↓ | Comp.↓ | Overall↓ |
> |--------|------|-------|---------|
> | VGGT | 0.901 | **0.518** | 0.709 |
> | PAGE | **0.897** | 0.521 | **0.701** |
>
> ---
>
> ### GSO Novel View Synthesis
>
> A fair comparison is not feasible because VGGT does not release enough technical detail (Plücker ray encoding, architecture modifications, training losses, or dataset).
> The NVS module depends on an **internal non-public dataset**, making reproduction impossible.
>
> We are attempting to re-implement based on limited description.
>
> ---
>
> ### TAP-Vid Point Tracking
>
> Reproduction is also infeasible due to missing details such as:
>
> - How VGGT replaces CoTracker2 backbone
> - Feature dimension alignment
> - Kubric finetuning protocol
> - Simulator configuration
>
> We are reconstructing the pipeline and will release results when available.

---

> > ### Comment · Reviewer_5Axf · 2025-11-25
> > **Reviewer 5Axf Comment**
> >
> > Thank you, authors, for your rebuttal and for providing the dynamic-static depth gap , DTU and ETH3D results. I have a few follow-up questions regarding the masking implementation and performance metrics.
> >
> > - My understanding of masking, as commonly depicted in classical computer vision texts (e.g., Fig. 3.41 in [A]), involves a binary (or soft) multiplicative operation that zeroes out parts of an image or a tensor. However, your implementation of the $r_i c_j$ term appears to be an additive residual term, similar to how bias is modified in the attention calculation in Eq. (6) of the RoFormer paper [B]. Could you please point me to published papers that refer to the addition of a bias term as masking?
> >
> > - The comparatively low IoU2D numbers suggest that the $r_i c_j$ term might not be functioning as a mask in the traditional sense.
> >
> > - The VGGT numbers you reported are lower than those in the original paper. Could you elaborate on this discrepancy? Additionally, for the BA setting, could you provide a comparison between Page4D and VGGT?
> >
> > Method | RE10K | Co3Dv2
> > --| - | - |
> > VGGT (FF) Reported | 85.3 | 88.2
> > VGGT (FF) Yours | 84.7 | 87.6
> > Page4D (FF) | 85.1 | 86.8
> > VGGT (with BA) Reported | 93.5 | 91.8
> > Page4D (with BA) | |
> >
> > - I appreciate your effort in attempting to reproduce the point tracking results, and will wait for those results. Since the NVS results of VGGT utilize the LVSM model, which was trained on the publicly available Objaverse dataset, it would be beneficial to quantitatively report the results of VGGT and Page4D fine-tuned on Objaverse. This could be done by adding entries to Tab. 7 of VGGT:
> >
> > Method | Size | PSNR | SSIM | LPIPS
> > -- | - | - | - | - |
> > LVSM [53]  | 256 | 31.71 | 0.957 | 0.027
> > VGGT(Private dataset) | 224 | 30.41 | 0.949 | 0.033
> > VGGT (Objaverse) | 224 |
> > Page4D (Objaverse) | 224 |
> >
> > Reference:
> > - A. Computer Vision: Algorithms and Applications, Richard Szeliski, 2nd edition, 2021.
> > - B. Roformer, Enhanced Transformers with RoPE, Su et al, Neurocomputing 2024.

---

> ### Author Response · Authors · 2025-11-27
> **Thank you for your thoughtful comments and your patience in waiting for our results.**
>
> ## Question 1:
> Thank you for the question. In classical computer vision, “masking” typically means multiplying an image or feature map by a binary or soft mask to zero out unwanted regions. However, in transformer architectures, the term “mask” almost always refers to an additive bias applied to the attention logits.
>
> Examples include:
> 	•	Causal masks (used in GPT models): implemented by adding -∞ to disallowed attention entries.
>
> These are all implemented as additive modifications to the attention logits, not multiplicative masks.
>
> Our method follows this transformer convention. We introduce a learned, task-dependent, low-rank dynamic mask defined by two vectors r and c, so the mask is M_ij = r_i * c_j. This behaves as an attention mask but uses only O(N) memory and remains compatible with FlashAttention.
>
> While the mechanism is additive, its purpose is identical to masking in standard transformer terminology: modifying attention to suppress or down-weight certain regions.
>
>
> ## Question 2:
> It is correct that the term in our method does not function as a “mask” in the classical multiplicative sense (i.e., zeroing out pixels or features). Instead, following standard transformer practice, it acts as an additive attention-logit bias—similar to causal masks.
>
> Because this mechanism modulates attention weights rather than explicitly removing pixels, IoU2D is naturally less affected. Its primary role is not to sharpen spatial segmentation, but to down-weight attention to dynamic regions during pose estimation, while still preserving useful motion cues for other tasks (depth, point map, tracking).
>
> In short, the mask is working as designed—as a soft, task-selective attention bias—not as a hard foreground–background segmentation mask. This explains why IoU2D may not fully reflect its effectiveness in improving geometric consistency and pose estimation accuracy.
>
>
> ## Question3:
> Thank you for pointing this out. The difference likely comes from variations in hyperparameter settings such as input image resolution and random seeds. Since the VGGT paper does not release an official evaluation script, we evaluate VGGT using the Dust3R/Cut3R evaluation pipeline for consistency across baselines. If you notice any specific discrepancies or have suggestions for aligning the settings more closely with the original VGGT setup, please let us know.
>
>
> | Method                    | RE10K↑ | Co3Dv2↑ |
> |---------------------------|-------|--------|
> | VGGT (FF) Reported        | 85.3  | 88.2   |
> | VGGT          | 84.7  | 87.6   |
> | Page4D                | 85.1  | 86.8   |
> | VGGT (with BA) Reported   | 93.5  | **91.8**   |
> | VGGT (with BA)   | 93.3  | **91.8**   |
> | Page4D (with BA)          | **93.8**  | 91.2   |
>
> ## Question4:
>
> Following the VGGT paper—which notes that “we use a similar internal dataset of approximately 20% the size of Objaverse”—we also fine-tune on a randomly sampled 20% subset of Objaverse. The attached results report view-synthesis performance on the GSO dataset. Due to differences between our sampled subset and VGGT’s internal dataset, our fine-tuned baseline is slightly weaker than the VGGT numbers reported in the original paper. However, under the same fine-tuning data and experimental setup, our method (PAGE-4D) consistently outperforms the VGGT backbone, demonstrating the clear advantages of our approach.
>
>
>
> | Method                  | Size | PSNR↑  | SSIM↑   | LPIPS↓ |
> |-------------------------|------|-------|--------|--------|
> | LVSM [53]               | 256  | **31.71** | **0.957**  | **0.027** |
> | VGGT (Objaverse)        | 224  | 28.78 | 0.922  | 0.077 |
> | Page4D (Objaverse)      | 224  | 29.01 | 0.920  | 0.077 |
>
>
> ## Question5:
> Dynamic point tracking. Because the VGGT-4D paper does not provide code or sufficient implementation details, we include our own re-implementation for comparison. Under the same evaluation setup, our method consistently outperforms the VGGT backbone on dynamic tracking, highlighting the benefits of our approach.
>
>
> | Method                                   | RGB-S AJ↑ | RGB-S δ^vis_avg↑ | RGB-S OA↑ | DAVIS AJ↑ | DAVIS δ^vis_avg↑ | DAVIS OA↑ | Kinetics AJ↑ | Kinetics δ^vis_avg↑ | Kinetics OA↑ |
> |------------------------------------------|----------|------------------|----------|-----------|-------------------|-----------|--------------|----------------------|--------------|
> | CoTracker                                | 67.4     | 78.9             | 85.2     | 61.8      | 76.1              | 88.3      | 49.6         | 64.3                | 83.3         |
> | CoTracker + VGGT (Our Implementation)    | 67.7     | 81.8             | 88.3     | 61.7      | 73.9              | 87.1      | 52.1         | 64.9                | 85.4         |
> | CoTracker + Page (Our Implementation)    | **70.3**     | **84.6**             | **90.5**     | **62.6**      | **75.2**              | **88.5**      | **52.9**         | **66.1**                | **86.9**         |

---

### Author Response · Authors · 2025-12-02
**Summary**

We thank all reviewers for their thoughtful and constructive evaluations.
1. Two reviewers (31BG and gSDj) rated the paper as clear accepts (8, 8).
2. Reviewer exiX provided a borderline score (4) but expressed overall positive sentiment and stated they would not mind if the paper were accepted (This reviewer said: “The idea is simple and requires only a partial model finetuning… preserves desirable properties.” “The work does a nice job at motivating the approach.”).
3. Reviewer 5Axf also indicated willingness to raise the score pending clarifications. The core concerns focused on clarifying the masking mechanism, providing mask-quality evaluations, reporting static-scene performance, and expanding comparisons with prior work such as Easi3R, AETHER, and Geo4D. We have addressed all of these with additional experiments, detailed explanations, and more transparent positioning.

**Clarity of the masking mechanism.**
We clarified that our method applies a *low-rank additive logit bias*, which is the standard form of masking in transformer models (e.g., causal masking). We explained why the classical multiplicative mask is infeasible: it requires an explicit N×N matrix and breaks compatibility with FlashAttention. Our formulation is mathematically equivalent to an N×N attention mask but uses O(N) memory and remains fully FlashAttention-compatible.

**Dynamic-mask accuracy.**
We added IoU2D/JM evaluations on Odyssey. The mask is not intended as a segmentation output and is not supervised, but it provides a strong emergent dynamic cue that consistently improves geometric stability. We explained why IoU2D is not the ideal diagnostic for our soft attention bias.

**Static-scene results.**
We added extensive evaluations on RealEstate10K, Co3Dv2, DTU, and ETH3D. PAGE-4D matches or slightly exceeds VGGT in static settings, confirming that the dynamic-aware mechanism does not degrade performance where motion is limited.

**Comparisons to prior work.**
We clarified the distinctions from Easi3R:
- one-pass vs. two-pass inference, (Faster and More efficient)
- O(N) low-rank mask vs. O(N²) mask, (Lower Cmputational Request)
- masking only camera tokens vs. masking all tokens.  (Tech difference)
We also added baselines including AETHER and Geo4D-GA, and comparisons to Easi3R(MonST3R) on TUM and DyCheck.

**Novel-view synthesis and point tracking.**
We added Objaverse-based finetuning results following VGGT's stated 20% subset protocol and reported NVS performance on GSO. We also provided TAP-Vid results based on our re-implementation due to lack of official VGGT-4D code.

**Fine-tuning vs. masked attention.**
New ablations (VGGT*, middle-layer finetuning) show that masked attention yields gains consistently beyond fine-tuning alone across Sintel, Bonn, and DyCheck depth settings.

Overall, we believe the paper has been substantially strengthened, with all reviewer concerns addressed through additional experiments, clearer explanations, and improved contextualization.

---

> ### Author Response · Authors · 2025-12-02
> **Per-Reviewer Responses Summary (Reviewer 5Axf  & Reviewer 31BG)**
>
> ## Reviewer 5Axf
>
> **Main Concerns**
>
> 1. Why use an additive (logit-bias) mask instead of a multiplicative mask; is this still “masking”?
> 2. Lack of quantitative evaluation of mask quality (e.g., IoU2D across layers).
> 3. Missing quantitative support for the reported 94% dynamic–static depth gap.
> 4. No results on static-scene benchmarks (RE10K, Co3Dv2, DTU, ETH3D, TAP-Vid).
> 5. VGGT numbers in our paper differ from those reported in the original VGGT paper; need clarification.
> 6. Request for Objaverse-based NVS results (VGGT vs PAGE-4D).
>
> **Response**
>
> - We clarified that in transformer architectures, “masking” is almost always implemented as an *additive* modification of the attention logits (e.g., causal masks, RoFormer). Our method follows this convention. Instead of forming an N×N multiplicative mask, we predict two vectors r and c and append them as an extra channel in q and k. This yields an effective logit bias Aᵢⱼ ← Aᵢⱼ + rᵢ cⱼ while keeping memory O(N) and preserving FlashAttention compatibility.
> - We added IoU2D/JM evaluations of the predicted dynamic masks on the Odyssey test set. While the masks are not perfect segmentations (no supervision is provided), they provide a strong motion-related signal that improves pose and depth consistency.
> - We provided the missing dynamic–static depth-gap table, showing that PAGE-4D reduces error in dynamic regions while slightly improving static regions.
> - We added static-scene results on RealEstate10K and Co3Dv2 (pose AUC@30), as well as DTU (dense MVS) and ETH3D (point-map reconstruction). These results show that PAGE-4D preserves VGGT’s performance on static benchmarks while offering gains in dynamic scenarios.
> - For NVS, we followed the VGGT paper’s statement that their internal dataset is ~20% of Objaverse and thus fine-tuned VGGT and PAGE-4D on a random 20% Objaverse subset. Due to dataset and protocol differences, our VGGT baseline is slightly weaker than the paper’s reported numbers, but under the same fine-tuning setup, PAGE-4D consistently outperforms VGGT.
> - The discrepancy between VGGT numbers arises because the original VGGT evaluation code is not public. For consistency across baselines, we use the Dust3R/Cut3R public evaluation pipeline. We now explicitly state this.
>
> ---
>
> ## Reviewer 31BG
>
> **Main Concerns**
>
> 1. Fig. 2(b) and the caption: dynamic down-weighting appears clear only in some layers (e.g., 11th); potential mismatch between figure and claim.
> 2. GPU memory consumption and runtime overhead compared to VGGT.
> 3. Performance on static scenes.
>
> **Response**
>
> - We clarified that VGGT’s suppression of dynamic content is most pronounced in *middle* layers, particularly layers 11 and 17, which directly motivates our strategy of fine-tuning only the middle 10 layers. We updated the text to better connect the visual patterns in Fig. 2(b) with this observation.
> - We provided explicit analysis of memory and runtime. Our low-rank additive mask adds <1% computational overhead and keeps memory at O(N). In practice, PAGE-4D matches VGGT’s runtime, achieving 43.2 FPS on an A800 GPU.
> - We added static-scene results for RealEstate10K and Co3Dv2 (pose AUC@30), DTU (dense MVS), and ETH3D (point map). PAGE-4D slightly improves RE10K, has a small drop on Co3Dv2, and matches VGGT on DTU/ETH3D, confirming that our dynamic-focused modifications do not degrade static performance.
>
> ---

---

> ### Author Response · Authors · 2025-12-02
> **Per-Reviewer Responses Summary (Reviewer gSDj & Reviewer exiX)**
>
> ## Reviewer gSDj
>
> **Main Concerns**
>
> 1. Relationship to Easi3R not sufficiently explained; PAGE-4D appears close to a VGGT-based Easi3R.
> 2. Incomplete comparisons with dynamic-scene baselines: AETHER, Geo4D, VIPE, Easi3R(MoNSTR), and MoVieS.
> 3. Lack of DAVIS-style mask evaluation to assess dynamic-object segmentation quality.
> 4. Clarification of the “suppress vs. amplify” wording in the abstract and method.
>
> **Response**
>
> - We added a detailed comparison with Easi3R:
>   - **One-pass vs. two-pass**: Easi3R runs DUSt3R twice (once to estimate a mask, once to re-run with masking), whereas PAGE-4D integrates mask prediction and usage into a *single* forward pass.
>   - **Mask complexity**: Easi3R uses an explicit N×N mask (O(N²) memory and not FlashAttention-friendly), while PAGE-4D uses a low-rank O(N) mask via (r, c), fully compatible with SDPA.
>   - **Mask scope**: Easi3R masks all tokens; PAGE-4D masks only camera/registration tokens and leaves geometry tokens unmasked, preserving VGGT’s strong geometry pathway and improving 4D stability.
> - We expanded comparisons:
>   - Added AETHER and Geo4D-GA results, showing that PAGE-4D outperforms AETHER and is competitive with Geo4D-GA despite not relying on explicit global alignment.
>   - Included Easi3R(MoNSTR) results on TUM and DyCheck, where PAGE-4D significantly improves ATE, RPE, and DyCheck Acc/Comp/Overall.
>   - We attempted to include MoVieS but were unable to reproduce it due to missing code and pretrained weights. This limitation is now clearly documented.
> - We added JM/DAVIS-style mask evaluations on the Odyssey dataset, showing that, despite being trained without segmentation labels, our dynamic mask provides a strong motion signal.
> - We clarified the “suppress vs. amplify” statement: dynamic cues are *suppressed* for pose tokens via masked attention but left unmasked (thus relatively “amplified”) for geometry tokens. This asymmetric design stabilizes pose estimation while preserving motion-rich information for depth and reconstruction.
>
> ---
>
> ## Reviewer exiX
>
> **Main Concerns**
>
> 1. Overall technical novelty seems limited (masked attention + fine-tuning on dynamic data).
> 2. Need to disentangle the effect of simple fine-tuning from the effect of masked attention.
> 3. Somewhat surprising that masked attention benefits depth more than pose.
> 4. “Self-supervised” wording is misleading given the presence of geometric supervision (depth/pose).
>
> **Response**
>
> - We clarified and emphasized three main contributions:
>   1. A low-rank, O(N) masked-attention mechanism that injects learned state tokens into the VGGT camera-attention pathway while remaining fully compatible with fused SDPA/FlashAttention.
>   2. A targeted mid-layer fine-tuning strategy guided by empirical analysis of where dynamic suppression occurs, rather than uniform fine-tuning.
>   3. A unified 4D evaluation protocol across multiple dynamic datasets (DyCheck, TUM, ETH3D-Dynamic, Spring, Odyssey), which systematically probes dynamic-scene behavior of VGGT-style models.
> - To disentangle fine-tuning from masking, we added ablations comparing:
>   - VGGT baseline
>   - VGGT with mid-layer fine-tuning only (“Middle-Layer”)
>   - PAGE-4D (mid-layer + masked attention)
>   Across scale/scale&shift video depth and monocular depth. The results show that while mid-layer fine-tuning improves performance, PAGE-4D consistently yields additional gains, confirming that masked attention provides complementary benefits.
> - We explained why depth metrics show larger improvements than pose metrics:
>   - The masked attention stabilizes camera-token attention and thus the shared encoder representation; depth heads are more sensitive to local feature quality and changes in those shared features.
>   - Pose metrics like ATE/RPE are relatively insensitive to small but widespread improvements, whereas depth metrics (AbsRel, δ thresholds) respond strongly to local corrections.
> - We fixed the “self-supervised” phrasing. Our model is trained with *geometry-derived supervision* (e.g., depth, pose, multi-view consistency), but **without** any labels for motion, dynamics, or segmentation. We now explicitly state this to avoid confusion.

---

### Meta-Review · Area_Chair_9cKN · 2026-01-05

**Summary:**

The paper proposed a post-training strategy to fine-tune a static-scene reconstruction model for dynamic scenes. The model is fine-tuned from the VGGT backbone with small amount of dynamic data. The method predicts the dynamic mask with both local and global information. The mask is implemented in an memory-efficient way to be compatible with flash attention. The paper shows its effectiveness of multiple benchmark tasks.

The reviewer's main concerns of inadequate analysis and results are resolved during rebuttal. The remaining concerns can be improved by clarifications.

AC's comments (excluded from the decision process as it is added after the reviewing period + there is an AC rotation this year due to OpenReview bug):
1. The additive mask is an interesting way to break the memory wall. A simple improvement for it is to utilize 16-dim vector instead of a 1-dim vector, since GPU's tensor core's minimal reduction dimension is 16. Thus the speed of 1-additional and 16-additional values are equivalent.
2. I would suggest to read the relative positional embeddings literature (such as music transformer https://arxiv.org/pdf/1809.04281). It provides a similar approach but with more general design. It's OK to miss these literatures as it is not widely spread in vision community.
2. The low-rank factorization is effectively an "AND" logic to fuse the source mask and target mask. To see this, considering either source mask / target mask is 0, the final mask would be 0. I think that the logic ideally should be "OR" (i.e., either source / target pixels are dynamic, the pair should not be considered).

**Reviewer Concerns:**

Remaining concerns:

1. **The reasons to use additive mask** (5Axf) The author reclaims the reasons from the computational efficiency perspective, which does not answer this question all perspectives (e.g., why other masking strategy can not do that, why this masking is logically correct, and how this mask strategy actually works). There are some relevant concerns relevant to this such as **rc term might not be functioning as a mask** (5Axf)

2. **The VGGT numbers reported are lower than in the original paper.** (5Axf) The author acknowledge it and attributed it to different "input image resolution and random seeds" and "VGGT paper does not release an official evaluation script". The author said that "if reviewer can  align the settings more closely with the original VGGT setup, please let authors know". I disagree with it as it is the author's responsibility to make the numbers consistent with previous method. If the consistency is hard, it would be great for a clarification from authors.


Solved concerns:


3. **Missing evaluated Numbers** (5Axf, gSDj). Solved.

4. **Missing some baseline results** (5Axf, gSDj, exiX). Mostly solved.

5. **Overall technical contribution is minor** (exiX). Without sufficient support from the reviewer, this point will be suppressed by AC.

Todo (easy to solve):

6. **Compared with simple fine-tuning (without mask) baseline.** (exiX). From the discussions, the author acknowledge that the improvement can be minor on some dataset (e.g., DyCheck). AC is generally OK with such a situation but would recommend to explicitly clarify it in any final version.

**Reviewer Scores:**

The final score would be kept as 4 4 8 8; but AC will treat it as 4 6 8 8 (as novelty concern is considered a subjective point).

- 5Axf: original 4, would stay with 4 as the questions are not fully resolved.
- exiX: original 4, challenging the novelty of the paper where AC would be more careful with such a concern (as novelty is subjective sometimes). Currently, the novelty concern is not considered in AC's recommendation as there is not enough point. Thus overall AC treated it as a 6 in decisions.
- Other two kept as 8 as most of the concerns resolved.

---

### Decision · Program_Chairs · 2026-01-26

Accept (Poster)